# Functional Modules in the Meristems: “Tinkering” in Action

**DOI:** 10.3390/plants12203661

**Published:** 2023-10-23

**Authors:** Ksenia Kuznetsova, Elena Efremova, Irina Dodueva, Maria Lebedeva, Ludmila Lutova

**Affiliations:** Department of Genetics and Biotechnology, Saint Petersburg State University, Universitetskaya Emb. 7/9, 199034 Saint Petersburg, Russia; kskuz95@mail.ru (K.K.); efremova.bio@gmail.com (E.E.); m.a.lebedeva@spbu.ru (M.L.); l.lutova@spbu.ru (L.L.)

**Keywords:** land plants, functional module, meristem, phase transition, peptide/protein ligand, regulation of transcription, evo-devo

## Abstract

Background: A feature of higher plants is the modular principle of body organisation. One of these conservative morphological modules that regulate plant growth, histogenesis and organogenesis is meristems—structures that contain pools of stem cells and are generally organised according to a common principle. Basic content: The development of meristems is under the regulation of molecular modules that contain conservative interacting components and modulate the expression of target genes depending on the developmental context. In this review, we focus on two molecular modules that act in different types of meristems. The WOX-CLAVATA module, which includes the peptide ligand, its receptor and the target transcription factor, is responsible for the formation and control of the activity of all meristem types studied, but it has its own peculiarities in different meristems. Another regulatory module is the so-called florigen-activated complex, which is responsible for the phase transition in the shoot vegetative meristem (e.g., from the vegetative shoot apical meristem to the inflorescence meristem). Conclusions: The review considers the composition and functions of these two functional modules in different developmental programmes, as well as their appearance, evolution and use in plant breeding.

## 1. Introduction

In biology, a functional module is a conserved complex of interacting components that can perform a function in different parts of the organism in a nearly unaltered form. As part of a more complex system (organism), the functional modules are universal, functionally equivalent, interchangeable and relatively autonomous. The systemic regulation and evolution of the modules, as parts of a complete system, is carried out by changing the rate of formation and reducing or increasing the number of modules or changing their structure and composition, forming new elements and new relationships [1,2].

The idea of creating a variety of life forms by combining certain parts has been “in the air” since ancient times: it goes back to Empedocles, who believed that when different parts of animals first appeared, then they were formed into various combinations, with non-viable variants dying and successful ones surviving [3]. In a more mature form, the principle of such “bottom-up” block evolution was formulated by François Jacob [4], who proposed the idea of “tinkering”—a combination of constituent parts at both the molecular and organismal levels. According to this idea, complex structures are much less likely to emerge de novo than they arise by modifying existing systems or structures: as one of the most interesting manifestations of tinkering, Jacob described the complication of the nervous system and the emergence of the human brain. Thus, in the course of evolution, especially in macroevolutionary events, some universal constructions are used from time to time, which are “approved” in the early stages of evolution, and then replicated and used by a variety of organisms, which serves as an excellent illustration of one of the fundamental principles of biology—the principle of biological universality [5].

There are modules at different levels of organisation of biological systems, e.g., morphological and molecular. The plant body consists of several modules at different levels of organisation. Our review will focus on the plant meristems as morphological modules, and on the molecular modules that underlie the control of meristem homeostasis.

At the morphological level, the modules are usually unified, equivalent and interchangeable morphofunctional subsystems which are formed as a result of structural and functional differentiation. There are two fundamentally different forms of body organisation in multicellular organisms—modular and unitary: they are found in all kingdoms of the living organisms, but are usually illustrated by the examples of higher plants and vertebrates, correspondingly. Modular organisation is usually defined by the capability for open growth and cyclic morphogenesis: modular organisms, such as higher plants, are able to reproduce their elements repeatedly throughout their lifetime [1]. In the case of higher plants, this ability arose as a consequence of the sessile lifestyle and the inability to actively avoid adverse environmental conditions, resulting in the necessary acquisition of high developmental plasticity and high regenerative capacity, which are successfully achieved through the modular structure of the body.

The basis of this principle of modular organisation of the plant body is based on the existence of meristems, which are specialised structures containing stem cell (SC) niches in which histogenesis and organogenesis take place throughout the entire postembryonic plant development. Unlike animal SCs, totipotent plant SCs persist in meristems for a long time and support organogenesis throughout the life cycle. The meristems of higher plants are generally designed according to a common plan. Typical plant meristems, in particular, shoot and root AMs (SAM and RAM, respectively) and the LM (pro)cambium, contain SCs in the central zone, while specialised cell types are differentiated at the periphery. Thus, two opposing processes are at the core of meristem activity: self-renewal of SCs in the central part of the meristem and differentiation of specialised cells at the periphery. A specific feature of at least the AMs and the cambium is the presence of the OC, which is a small group of slowly dividing cells (or a layer of cells in the cambium) that acts as a source of the signal that inhibits the differentiation of adjacent SCs. The cells of the OC have the ability to divide asymmetrically, with one of the daughter cells remaining in the OC and the other crossing its borders to enter the differentiation pathway [6,7]. The inability of the OC to function properly, for example, as a result of the mutations in the key regulatory genes that support OC identity, leads to the loss of the meristem’s ability to maintain itself, and its activity rapidly ceases [8,9,10]. The cells of OCs are maximally similar to animal SCs in terms of their properties, including location (always in the same position), a slow division, lack of differentiation marks and the ability to divide asymmetrically [7,11]. Thus, higher plant meristems are one of the clearest examples of functional modules at the organismal level, and in the remainder of this review, we will focus mainly on the molecular modules that regulate meristem development and function.

The assembly and function of morphological modules are based on the activity of molecular modules containing specific players, such as receptors for different ligands, regulators of signal transduction, transcription factors (TFs), target genes and their products, etc.

The reason for the formation of conservative functional modules with approximately the same “composition of participants” is the presence of conservative protein or nucleic acid sequences used for the interaction of the components in the module. In this respect, several types of functional modules with different compositions could be distinguished (see Figure 1). For example, a functional model could include TFs, their cofactors and target genes (e.g., interacting TFs of the KNOX and BELL families in the regulation of *IPT* genes [12] or WUSCHEL-RELATED HOMEOBOX (WOX) and HAIRY MERISTEM (HAM) TFs in the regulation of genes encoding CLAVATA3 (CLV3)/Embryo Surrounding Region-Related (CLE) peptides [13]). Such modules arise due to the conservatism of TF binding sites in the promoters of target genes and also of the domains responsible for protein–protein interactions of partner TFs. Secondly, a functional model could comprise a ligand and its receptor (sometimes with co-receptor) and the underlying signalling pathway (such as the peptide hormones of several families whose receptors are leucine-rich repeat receptor-like protein kinases, LRR-RLKs). These modules are formed by the structural “correspondence” of receptors and ligands [14]. Thirdly, a functional model could include small RNAs and their targets [15]. This module (which is beyond the scope of this review) is based on the complementary interaction of small RNA sequences with their target transcripts. The examples of such small RNA-based modules include conserved miRNA156 and its target transcripts of *SQUAMOSA PROMOTER BINDING PROTEIN-LIKE* (*SPL*) genes, and miRNA172 and transcripts of *APETALA2* (*AP2*)-like genes, which are key regulators of the flowering programme in plants [16], and miRNA165/166 targeting the transcripts of the *HD-ZipIII* gene, which are well-known regulators of SAM formation, organ polarity and vascular development [17] (see Figure 1).

Simple modules can sometimes give rise to complex ones. A classic example of a complex module in plants is the WOX-CLAVATA system, which includes the mobile signalling peptides CLE, their receptors, LRR-RLKs, and downstream target genes encoding WOX family TFs, some of which can regulate the *CLE* gene expression, forming a negative regulatory loop [18,19]. WOX-CLAVATA systems are known to regulate the development of different types of meristems, early embryogenesis, response to external signals (e.g., soil nitrate levels) and even interaction with symbionts and pathogens (reviewed in [20,21]).

Our review is devoted to describing the best known examples of regulatory modules that control meristem activity and reorganisation in higher plants. Numerous studies have shown that many plant developmental programmes (e.g., control of meristem activity, transition to flowering, formation of certain specialised cell types, etc.) are controlled by molecular modules that are highly conserved and often used in different developmental programmes. For example, the WOX-CLAVATA systems mentioned above are so versatile and conservatively designed that it makes sense to look for a CLE peptide that regulates WOX wherever it has been shown to work. Another example of conservative regulatory modules in plants is the florigen-activating complex (FAC), which includes the FLOWERING LOCUS T (FT)-like mobile proteins, their receptors from the 14-3-3 protein family, and the FLOWERING LOCUS D (FD) TF that they regulate. Thus, FAC is a hexameric complex composed of two FT molecules, two 14-3-3 molecules and two FD molecules; this complex is assembled directly upon binding to the promoters of target genes (mainly encoding TFs of the MADS family). Such a FAC module was at first identified as a central regulator of the transition to flowering, and then its involvement in the development of storage organs—potato tubers and *Liliaceae* bulbs—and in the control of the dormancy period of axillary buds was revealed (reviewed in [22]).

The strong functional link between the components of the regulatory module means that the changes in individual components of the functional module cause changes in the functioning of the whole module. The successful variants of such mutations can be fixed in evolution and create a new developmental programme. A striking example of such block evolution is the emergence of “antiflorigens”—TERMINAL FLOWER 1 (TFL1)-like proteins that differ from FT in some conserved amino acid residues, but can reverse the functions of entire FACs [23]. In addition to meristems, there are examples where mutations in the components of regulatory modules, leading to functional changes in the operation of the module as a whole, have been used by humans in the selection of agricultural crops, such as the use of genes regulating the transition from vegetative SAM to floral in the tomato breeding [24], or the selection of forms with increased expression of cambial regulators such as *WOX4* and *KN1* (*KNOTTED-LIKE HOMEOBOX 1*) in *Ipomea batata* compared to related species without storage roots [25].

The review will provide a clear example of the conserved functional modules in plant development at the morphological and molecular level—meristems, which to a first approximation consist of a set of similar components, and their conserved regulators, which function in the conserved interactions with each other. The examples of changes in the components of such modules underlie a number of evolutionary changes and sometimes of plant domestication and breeding.

## 2. Meristems—The Functional Module at the Level of the Organism

The existence of meristems enables plants to produce new organs (leaves, flowers, roots, etc.) throughout their life cycle. SAM and RAM produce the above- and below-ground organs, respectively, thus ensuring growth along the vertical axis, whereas lateral meristems, such as the (pro)cambium, are necessary for growth by thickening and forming the vascular system.

Meristems can be divided into primary and secondary meristems. Primary meristems are formed during the embryonic development and include primary apical meristems (SAM and RAM) and lateral meristems, the procambium and pericycle. During the postembryonic period, secondary meristems can arise from both primary meristems (for example, RAMs of lateral roots are formed from pericycle cells, and secondary cambium from the pericycle and procambium) or from specialised tissues as a result of dedifferentiation (for example, axillary meristems, a type of secondary SAM, are initiated from the leaf axil cells that retain meristematic characteristics). There are also additional or “facultative” meristems that occur in certain plant groups under certain conditions (such as nodule meristems in legumes or meristem-like structures such as wound callus, or galls and tumours formed in response to certain groups of pathogens), and some of these can also give rise to specialised organs. 

Despite the differences in structure, at least some of the meristems are organised according to a single principle, which is based on similar regulatory mechanisms involving molecular modules. Each meristem contains a pool of SCs in its central zone. Their daughter cell can remain in the central zone of the meristem to renew the SC pool, or it can move to the periphery of the meristem, where it starts the differentiation pathway to be incorporated into differentiated tissues or organ primordia. At the same time, despite the similarity in the general principles of organisation, there are significant differences in the meristems, both in their specific contribution to the formation of the plant body and in the detailed structure of their regulatory mechanisms.

In the RAM, SCs, or initials, surround the OC (here called the quiescent centre, QC), and the QC separates the distal and proximal parts of the RAM, which give rise to the root cap and the root body, respectively. The proximal SCs in the RAM are arranged in ordered cell files that divide asymmetrically to give rise to the provascular tissues, endodermis, cortex, epidermis, and lateral root cap, creating the radial symmetry of the root. Columella SCs are located distal to the QC. The molecular mechanisms of SC homeostasis in the RAM have been studied primarily in the columella initials, as the degree of their differentiation can be easily assessed visually by the accumulation of starch granules [26]. Although the SAM and RAM are structurally distinct, the regulation of their SC maintenance is carried out by very closely related regulatory modules [27].

The vascular cambium is a lateral meristem that gives rise to xylem and phloem tissues and is essential for radial growth. The procambium is a primary lateral meristem that forms during embryogenesis as a component of the primary vasculature, and the cambium is a secondary meristem: at least in the root, it originates from the procambium and the xylem pole pericycle. Like the SAM and the RAM, the cambium contains the SC pool and the OC: the cambial organiser is the layer of xylem initials adjacent to the initiating cambium, which confers SC identity to its neighbouring cell by initiating cambial divisions [28,29]. Each cambial SC, which undergoes asymmetric periclinal divisions, produces both xylem and phloem progenitors. The progenitors divide symmetrically at the periphery of the cambial zone and then differentiate proximal or distal, giving rise to layers of secondary xylem cells towards the centre, and secondary phloem cells towards the periphery [28]. 

The homeostasis and activity of plant meristems are also maintained by phytohormones such as indole-3-acetic acid (IAA), cytokinins (CKs), gibberellins and certain peptide phytohormones such as CLE peptides [30,31]. Furthermore, the same hormones can have the opposite effects on different meristems, e.g., the opposite effects of IAA and CKs on SAM and RAM [32]. Internal factors include molecular modules that ensure meristem activity, and since all plant morphogenesis, and meristem activity in particular, is regulated by environmental factors, the function of these modules is closely intertwined with the phytohormonal regulation of meristems.

The molecular modules underlying the functioning of meristems are numerous and very diverse, so we have to focus on describing only the two most studied examples of molecular modules that ensure meristem functioning: WOX-CLAVATA, which is essential for meristem maintenance, and FAC, which ensures phase transitions in meristems (e.g., transition from vegetative SAM to floral meristem, formation of “dormant” axillary meristems, or induction of lateral growth by activation of lateral meristem). These types of molecular modules will be discussed in more detail in the following sections.

## 3. Molecular Module 1: The WOX-CLAVATA System

The WOX-CLAVATA system is a complex regulatory module that includes CLE peptides, their receptors, and the target of the signalling pathway they induce—genes encoding homeodomain TFs of the WOX family. This highly conserved module controls the SC maintenance, the size and homeostasis of various meristems. The composition of the module is roughly the same in different meristems, varying in the composition of specific participants, the components that interact with them, and the targets of the module [33] (Figure 2). 

### 3.1. The WOX-CLAVATA System in the SAM

The functions of the WOX-CLAVATA system were first described in the SAM using a series of mutants with opposite phenotypes affecting SAM activity, including the *clavata* (*clv1*, *clv2*, *clv3*) mutants, characterised by increased SAM size and delayed SAM termination, and also the *wuschel* (*wus*) mutant, characterised by a premature termination of SAM activity. Thus, the *CLV* genes have been described as negative regulators of SAM activity, and *WUS* has been described as a positive regulator [18]. The *WUS* gene encodes a WOX family TF [8]. Among the *CLV* genes, *CLV3* encodes a short signal peptide [34] belonging to the CLAVATA3 (CLV3)/Embryo Surrounding Region (CLE) family of peptide phytohormones, while *CLV1* encodes its receptor belonging to the LRR-RLK family [35]. These genes show different expression patterns in the SAM. The *WUS* gene is only expressed in the OC, *CLV1* is expressed in the OC and the adjacent cells, whereas *CLV3* expression is observed in the two upper cell layers, L1 and L2, above the OC. From there, the CLV3 peptide can migrate to the cell layers that are closer to the centre, restricting the expression of *WUS* [18]. The WUS TF in turn directly activates the *CLV3* gene expression, forming a negative feedback in the regulation of the OC [18,36]. The expression zone of the *CLV1* gene overlaps with that of *WUS*, but is broader. Thus, CLV1 forms a barrier on the way of the mobile CLV3 peptide and protects the OC from its inhibitory effect [37]. The *CLV1* gene is also a direct target of the WUS TF: WUS negatively regulates its expression [38]. In addition, CLV1 receptors could undergo CLV3-dependent trafficking from the plasma membrane to the vacuolar compartment, which is another way to regulate SAM activity [39].

Subsequently, similar components including CLE peptides, their receptors, and WOX family TFs, were found in the different meristems, where they perform similar functions. Indeed, each of the components of this system belongs to a large family of proteins that can form closely related modules: in *Arabidopsis*, for example, there are 32 genes encoding CLE peptides [40], fifteen genes encoding WOX TFs [41], and a family of CLV1-like receptors representing a rather large group of proteins [42]. The functions of certain proteins within each of these families may either partially overlap or have a narrow specialisation (see below for details).

The SAM also contains several receptors for CLE peptides, such as CLV2, which forms a complex with CORYNE (CRN) [43], RECEPTOR-LIKE PROTEIN KINASE 2 (RPK2) [44] and BARELY ANY MERISTEM 1-3 (BAM1-3) [45]. Among these, CLV1 is a master receptor of CLV3 which acts in the central zone of the SAM [18]. BAMs can also bind directly to the CLV3 peptide and regulate meristem size [46,47], whereas RPK2 and the CLV2-CRN receptor complexes are unlikely to bind directly to the CLV3 and can be considered as co-receptors for CLV1 and/or BAMs [46]. In addition, there are CLAVATA3 INSENSITIVE RECEPTOR KINASES (CIKs) LRR-RLKs which function as co-receptors for CLV1, CLV2/CRN and RPK2 and also play an important role in the regulation of CLV3-mediated SAM homeostasis [48]. Furthermore, CLV2-CRN and BAMs are referred to as “broader profile” receptors that function in different plant tissues to sense different CLEs [43,44,45,49,50]. 

In addition to *CLV3*, which is exclusively expressed in the SAM, many other *CLE* genes are also expressed in the *Arabidopsis* SAM [51]. Several *CLE*s are able to induce *wus*-like phenotypes when overexpressed [52], but their exact role in SAM development remains unknown. At the same time, *CLE40*, which has been shown to be a key regulator of RAM development (see below), is also expressed in the SAM periphery and promotes *WUS* expression via the BAM1 receptor, thereby increasing the number of SCs. Thus, two CLE peptide-dependent antagonistic pathways controlling *WUS* level provide additional opportunities to regulate the SAM size. At the same time, it has previously been shown that *CLE40* can substitute for the *CLV3* function and complement the *clv3* mutant phenotype in the SAM when it was expressed from the *CLV3* promoter [53]. This is probably due to the presence of a complementary spatial expression pattern of the genes that regulate these two alternative pathways, since *CLV3* and *CLV1* are expressed in partially overlapping domains in the SAM CZ, whereas *CLE40* and *BAM1* are expressed in the periphery [47].

In addition to the CLAVATA system, *WUS* expression can be regulated by other TFs and CK. CK signalling initiates *WUS* expression during axillary meristem formation and shoot regeneration from callus via a pathway involving the CK receptor ARABIDOPSIS HISTIDINE KINASE4 (AHK4) [54] and the *Arabidopsis* response regulators ARR1 and ARR2, key TFs of the CK response that directly activate the WUS expression [55]. The class III HOMEODOMAIN LEUCINE ZIPPER (HD-ZIP III) TFs PHABULOSA, PHAVOLUTA and REVOLUTA (PHB, PHV and REV) can interact with ARRs and bind to the WUS promoter [56]. Moreover, high levels of CK signalling in the rib meristem stabilise the WUS protein, whereas WUS is destabilised in the CK-deficient regions of the CZ (exact mechanisms are unknown) [57].

The expression of *WUS* is restricted to the OC cells by ROW1 (REPRESSOR OF WUS1), a PHD domain-containing protein that binds to H3K4me3 to regulate target gene transcription [58].

The signalling pathway which acts downstream of CLV3 perception by its receptors that regulates *WUS* expression is poorly understood. Downstream negative regulators of CLV signalling acting below CLV3 include the POLTERGEIST (POL) and POLTERGEIST-LIKE (PLL) 2C protein phosphatases with a nuclear localisation sequence [59], the α-subunit of a heterotrimeric G-protein complex [60], the small GTPase ROP [61] and the MAP kinase cascade [62].

The homeodomain-containing TF WUS has been shown to bind to DNA through three different motifs: a canonical TAAT (TAAT(G/C)(G/C)) motif for homeodomain TFs, a G-box-like TCACGTGA motif and a TGAA motif [36,38,63,64]. WUS can act as a transcriptional activator, but in the SAM it mainly acts as a repressor, and its repressive function requires the recruitment of TOPLESS family transcriptional corepressors that mediate the interaction with histone deacetylases [38,65]. 

Numerous targets of the WUS TF have been found in transcriptomic studies of plants with inducible *WUS* overexpression and/or by chromatin immunoprecipitation. The list of identified targets of WUS (Table 1) includes components of the WOX-CLAVATA pathway, genes that respond to CK biosynthesis and signalling, and also genes involved in the leaf development [36,38,63,64,66,67]. 

The *CLV3* gene, whose expression is strongly increased after 2 h of WUS induction [66], is the first identified and most studied direct target of WUS, forming a negative feedback loop with it [18]. However, CLV3 is the best known target of the WUS TF, whose feedback interaction with WUS is essential for SAM homeostasis and whose regulation by the WUS TF has been studied in detail. 

Since the *WUS* expression domain does not overlap with that of *CLV3*, it has been proposed that the WUS protein migrates across plasmodesmata into adjacent cells, where it activates *CLV3* gene transcription [36,68]. This migration of WUS depends on its C-terminal domains: the WUS box, which is required for its nuclear retention, and the EAR (ethylene-responsive element binding factor-associated amphiphilic repression) domain, which is involved in its nuclear export. 

WUS-mediated activation of *CLV3* is concentration dependent: at low levels of WUS, observed far from the OC, *CLV3* is activated, and at higher concentrations, WUS forms homodimers that are unable to activate *CLV3* transcription, rendering *CLV3* expression in the OC impossible [64,69]. Thus, the ability of WUS to form dimers results in concentration gradients of free active WUS and CLV3 proteins in the SAM. At the same time, there may be a negative feedback loop between WUS and CLE40 in the SAM: *CLE40* has been shown to be repressed in a WUS-dependent manner [47]. The gene encoding the CLV1 receptor is also a WUS target, but its expression is repressed by WUS [38,66].

In addition, WUS directly represses several genes: (1) genes encoding type A ARRs, which act as repressors of CK signalling, thereby increasing the response to CK in the SAM [38,65]; (2) *LONELY GUY 4* (*LOG4*), one of the CK biosynthesis genes [70]; (3) gene encoding ARF5/MONOPTEROS (MP), an auxin-responsive TF that affects leaf formation through positive regulation of *PIN-FORMED* (*PIN*) genes [71]; (4) a number of genes involved in leaf development such as *KANADI1* (*KAN1*), *KAN2*, *ASYM-METRIC LEAVES2* (*AS2*) and *YABBY3* (*YAB3*) [66].

To regulate *CLV3* expression, WUS has also been shown to form heterodimers with the GRAS domain TFs HAM1 and HAM2 [13,72]. The *HAM1* expression is localised to L3 (corpus cell layer in the SAM), where its product can form dimers with the WUS TF. According to the present model, the formation of WUS–HAM heterodimers could inhibit *CLV3* expression in the OC (as well as WUS homodimers), whereas the absence of HAM in L1 and L2 allows the induction of *CLV3* by monomers of the WUS TF [72]. 

Another partner of WUS is SHOOTMERISTEMLESS (STM), a class I KNOX family homeodomain TF, which is a central regulator of SAM formation during embryogenesis and SC division in the postembryonic SAM [73,74]. Interaction with STM has been shown to increase the ability of the WUS to bind to the *CLV3* promoter and is essential for *CLV3* expression in L1 and L2 [75]. 

Thus, the activity of the WUS monomers, homodimers and heterodimers with different partner TFs in regulating *CLV3* transcription provides the key mechanism for controlling SAM size [76].

### 3.2. The WOX-CLAVATA System in the RAM

There is a system in the RAM that is very similar to the one that regulates the balance of SAM SCs. The *WOX5* gene is expressed in the QC and promotes SC maintenance in a non-cell-autonomous manner just as *WUS* does in the SAM [77]. As in the case of WUS function in the homeostasis of the OC and the SCs of the CZ in the SAM, *WOX5* activity determines the status of the QC and columella initials. The *wox5-1* mutant phenotype is characterised by rapid differentiation of the QC and columella initials, whereas *WOX5* activation leads to the repression of columella cell differentiation and activation of their division [77,78]. At the same time, ectopic *WOX5* expression maintains columella SC differentiation resulting in the formation of supernumerary small and starchless undifferentiated cells in the columella root cap [19,77]. The *WOX5* promoter is active only in the QC, but the WOX5 protein can also be transported to the initial cells, in particular the columella SCs, and regulate its target genes (see below).

The *WUS* and *WOX5* genes which regulate SAM and RAM, respectively, can substitute each other and restore the normal phenotype in the loss-of-function mutants in the reciprocal substitution experiments in the OC and QC, when the *WUS* gene expression under the *WOX5* promoter was induced in the root of *wox5* mutants and vice versa [77]. At the same time, overexpression of *WUS* under the heat shock promoter in a wider region of the RAM induced shoot SC identity and also leaf and flower development, suggesting that *WUS* controls SAM identity [79]. Similar data were obtained in the experiment with the ectopic expression of *WOX5: WOX5* activity in the SAM disrupted shoot development by repressing shoot-related genes, in particular the leaf regulator *YABBY1* (*YAB1*). Thus, similar to WUS in the SAM, WOX5 in the RAM can regulate root identity by downregulating the expression of shoot regulators [80].

*WOX5* expression in the QC is mainly restricted by three components: ROW1, which represses its expression outside the QC, similarly to the control of *WUS* expression in the SAM [81]; auxin via the ARF10 and ARF16 TFs, which block *WOX5* expression in the distal meristem zone [82]; and the PLETHORA/AINTEGUMENTA-LIKE (PLT/AIL) TF family and its interacting partners, which stimulate *WOX5* expression but restrict it to the QC [83]. 

Auxin is not the only phytohormone affecting root development, but it plays a pivotal role as it is necessary and sufficient for root development [84]. The RAM SCs zone coincides with the auxin concentration maximum, which is formed by PIN-mediated polar auxin transport and local auxin production in the RAM [85,86]. The RAM phenotype of auxin response mutants has been shown to be very similar to that of plants with impaired *WOX5* expression, e.g., certain loss-of-function *arf* mutants and gain-of-function *axr3* mutants with a stable version of the auxin response inhibitor INDOLE-3-ACETIC ACID INDUCIBLE 17 (IAA17) have reduced columella SCs differentiation [82,87]. At the same time, local levels of auxin and WOX5 have opposite effects on the balance of the columella SCs: auxin promotes their differentiation, while the WOX5 maintains their undifferentiated state. Auxin-mediated differentiation of columella SCs requires the auxin response factors ARF10 and ARF16, whose activities restrict *WOX5* transcription to the QC, and the IAA17/AXR3 transcriptional repressor, which can promote *WOX5* expression in the QC by repressing the response to auxin [82]. In addition, *WOX5* expression can be induced by other WOX TFs: for example, the initiation of adventitious roots in *Arabidopsis* and radish requires the redundant activity of the WOX11 and WOX12 TFs, which directly activate WOX5 and WOX7, two genes essential for the establishment of a RAM [88]. 

Another regulator of WOX5 expression in the QC is the family of PLT TFs, which, together with auxin, play an important role in the specification and maintenance of the RAM SC niche [89,90]. The PLT-auxin positive feedback loop plays a central role in the control of the root SC niche. The expression of *PLT1* and *PLT2* in *Arabidopsis* primary and lateral roots depends on the activity of ARF5/MP and other ARF family TFs [91]. At the same time, *PLTs* are not primary auxin response genes because auxin turns on *PLT* transcription slowly, taking about 21 h to activate them [89]. Since the auxin-dependent expression of *PLTs* is delayed, the concentration gradients of auxin and PLT proteins coincide in space but not in time, and this effect could create a distance between the cell division zone and the QC and guide the progression of RAM cells from the SC state to the transit-amplifying cell state and finally to differentiation [91,92]. On the other hand, PLT TFs can moderate auxin levels and responses: PLT1 positively regulates the expression of *PIN* [90], *ARF5/MP* and certain *YUC* [93] genes. 

Recently, several PLTs have been shown to directly regulate *WOX5* expression in the QC, and PLTs have been shown to interact with WOX5 to regulate the expression of target genes (see below) [83,94]. The binding sites for PLT group TFs have been found in the *WOX5* promoter [93], and PLT3 has been shown to activate *WOX5* transcription [83]. In addition to regulating root SC specification and maintenance and *WOX5* expression, PLT TFs may interact with other key TF regulators of RAM [83]. These include SCARECROW (SCR), the GRAS family TF required for asymmetric division of cortex/endoderm initials [95], which also regulates QC maintenance through its cell-autonomous activity [96,97]. Another PLT-interacting TF is TEOSINTE-BRANCHED CYCLOIDEA PCNA (TCP), plant-specific TFs that play a role in SAM, RAM and leaf development by coordinating cell proliferation and differentiation [83]. The proteins PLT1, PLT3, SCR, TCP20 and TCP21 were found to work together to establish QC identity during embryogenesis, primary root development and lateral root formation. To do this, PLTs, SCR and TCP20 assemble into a complex in vivo where TCPs interact with PLTs and SCR through distinct regions [83]. Analysis of *WOX5* promoter activities, determined by measuring luciferase (LUC) intensities, showed that the combination of PLT3 with TCP20 and SCR increased *WOX5* promoter activity more than PLT3 alone, indicating that TCP20 and SCR are positive regulators of PLT-mediated *WOX5* induction [83].

The expression of *WOX5* in the RAM is controlled by the CLE40 peptide and its receptors. The CLE40, which is close to CLV3 and also acts in the SAM by antagonising CLV3 [47], is produced in the columella cells and promotes their differentiation [9]. The major receptor for CLE40 in the RAM is the receptor-like kinase ARABIDOPSIS CRINKLY4 (ACR4) [9], the only receptor for CLE peptides that does not belong to the LRR-RLK family and is a member of the CRINKLY4 (CR4) serine/threonine receptor-like kinase family [98]. Both *CLE40* and *ACR4* are expressed in the distal domain of root meristems, and *cle40* and *acr4* mutants have an increased elongation of columella SCs [9]. 

The *CLV1* gene, which encodes the major receptor that regulates SAM maintenance, is also expressed in the RAM immediately distal to the QC, partially overlapping the *ACR4* expression pattern, and also contributes to RAM SC control. In addition, CLV1 and ACR4 receptors can form homo- and heteromeric complexes with differential distribution at the plasma membrane and plasmodesmata [99]. Such CLV1-ACR4 complexes at plasmodesmata can bind secreted CLE40, and a “gating” model has been proposed in which the CLE40p/CLV1/ACR4 could control the mobility of the WOX5 protein to the distal RAM [100]. However, this model was not validated later [101]. 

In addition, many other CLEs can repress the RAM activity [40]: CLE14 and CLE20, for example, do so through their interaction with the CLV2/CRN receptors [40,102].

As with other WOX-CLAVATA systems, the cascade downstream of CLE40 and its receptors in the RAM has not been identified. At the same time, numerous experiments have provided extensive information about the targets of the WOX5 TF. These include genes whose products regulate the SC division and differentiation as well as phytohormonal homeostasis in the RAM (Table 1). 

Like WUS in the SAM, WOX5 negatively regulates SC differentiation in the RAM. This function of WOX5 is performed in different types of RAM initials, but has been studied mainly in columella SCs. The main target of the WOX5 TF in the columella SCs is the *CYCLING DOF FACTOR 4* (*CDF4*) gene, which encodes a group II DNA binding with one finger (Dof) TF [103], and regulates columella SC differentiation [19]. Thus, *CDF4* is normally expressed in the differentiating columella cells, but in the *wox5* mutants, its expression is increased and is also detected at the QC and near SCs, suggesting that WOX5 negatively regulates *CDF4*, thereby maintaining columella SC differentiation. The *CDF4* has been identified as a direct target of the WOX5 TF, which has been shown to bind to the TAAT motifs in its promoter region [19]. To repress *CDF4* transcription, WOX5 moves from the QC into the columella SCs, creating a CDF4 gradient opposite to the WOX5 gradient, thereby balancing SC maintenance and differentiation in the distal part of the RAM [19]. To regulate *CDF4* expression in columellar SCs, WOX5 interacts with TPL/TPR corepressors that recruit histone deacetylases [19]. This mechanism is very similar to that of WUS for its negatively regulated targets in the SAM [38,65,67].

However, it has recently been shown that the mobility of the WOX5 protein is not necessary to inhibit the differentiation of columellar SC [104]. Thus, WOX5 may act primarily in the QC where other short-range signals are generated that not only inhibit differentiation but also promote SC division in adjacent cells.

In the QC, WOX5 represses cell division by negatively regulating two genes encoding D-class cyclins (CYCD), which form complexes with cyclin-dependent kinase A (CDKA) to control the G1/S transition in the plant cell cycle through reversible phosphorylation of the RETINOBLASTOMA-RELATED protein. The WOX5-mediated repression of *CYCD* genes appears to be a key mechanism for establishing QC cell quiescence [105,106]. At the same time, outside the QC WOX5 stimulates the columella SC division through positive regulation of the *CYCB1;1* gene, which is known to be a regulator of cytokinesis [19,104,107]. 

**Table 1 plants-12-03661-t001:** The list of identified direct targets of the WUS and WOX5 transcription factors. (Upregulated target genes are highlighted in green, and downregulated target genes are highlighted in orange).

Transcription Factor	Target Gene	Encoding Protein and Its Functions	Reference
WUS	*CLV3*	Peptide phytohormone; negatively regulates *WUS* expression and size of SC pool in the SAM	[18,66]
	*CLE40*	Peptide phytohormone; positively regulates *WUS* expression and size of SC pool in the SAM	[47]
	*CLV1*	Receptor of CLE peptides in the SAM and RAM	[38,66]
	*ARR5*	Type-A ARR, repressors of CK signalling	[38,65]
	*ARR5*	Type-A ARR, repressors of CK signalling	[38,65]
	*ARR7*	Type-A ARR, repressors of CK signalling	[38,65]
	*ARR15*	Type-A ARR, repressors of CK signalling	[38,65]
	*LOG4*	Phosphoribohydrolase; catalyses the last step of CK biosynthesis	[70]
	*ARF5/MP*	Auxin-responsive TF; regulates the transport of IAA and the initiation of leaves and roots	[71]
	*AS2*	TF; regulates the initiation of leaf primordium	[66]
	*KAN1*	TF; regulates the development of abaxial side of leaf primordium	[66]
	*KAN2*	TF; regulates the development of abaxial side of leaf primordium	[66]
	*YAB3*	TF; regulates the development of abaxial side of leaf primordium	[66]
WOX5	*CDF4*	TF; regulates columella SC differentiation	[19,105]
	*CYCD1;1*	D-class cyclin; promotes the G1/S transition in the plant cell cycle	[105,106]
	*CYCD3;1*	D-class cyclin; promotes the G1/S transition in the plant cell cycle	[105,106]
	*TAA1*	Tryptophan aminotransferase; catalyses the first step of auxin biosynthesis via indol-3-pyruvate	[104]

Another group of WOX5 targets is involved in the regulation of auxin, a key hormone regulator of RAM maintenance and *WOX5* expression. WOX5 has been shown to regulate auxin peak formation in the RAM QC via the induction of auxin biosynthesis, and the direct target of the WOX5 TF here is the *TAA1* gene [104], which encodes tryptophan aminotransferase, an enzyme that catalyses the first step of auxin biosynthesis via indol-3-pyruvate (IPA). The second step of this biosynthetic pathway of biosynthesis is catalysed by flavin monooxygenases of the YUCCA (YUC) family [108], and a role for WOX5 in the expression of *YUCs* has also been proposed [87]. The expression domain of *TAA*, but not *YUCs*, overlaps strongly with that of *WOX5* [104], so that WOX5 can regulate the first step of auxin biosynthesis while maintaining the QC, but the regulation of the second step requires its movement. Thus, the maintenance of the columella SC depends on the WOX5-auxin feedback mechanism [82].

The WOX5 TF, like WUS, has “partner” TFs that enhance the interaction of WOX5 with target gene promoters—PLT3, which has also been shown to regulate *WOX5* gene expression [83,94]. The genetic interaction of PLTs and WOX5 is well established: they both act downstream of auxin and similarly promote root SC maintenance; *wox5 plt1 plt2* triple mutants show more pronounced defects in the RAM activity up to the complete meristem arrest [82]. Recent data indicate that PLT3 and WOX5 proteins can physically interact [83,94]. This interaction occurs differently in the QC and outside the QC. In the QC, PLT3 and WOX5 proteins are mainly located in the nucleoplasm, but in the nuclei of root SCs, PLT3 protein was mainly localised to the nuclear bodies (NBs) [94], which are subnuclear, membraneless, self-assembling protein/RNA-containing structures thought to function as a “nuclear dump” or “storage depot” [109]. Other PLTs and also WOX5 in the RAM SC nuclei are mostly located in the nucleoplasm, but namely the PLT3 protein can bind to WOX5 and recruit it to the NBs, thereby shortening its lifespan [94]. Thus, active WOX5 in the nucleoplasm may be required to maintain the stem status of QC cells, whereas its binding to PLT3 and compartmentalisation in the SCs may lead to a decrease in the stem status [94].

### 3.3. The WOX-CLAVATA System in the Cambium

In the (pro)cambium, the WOX-CLAVATA system controls the homeostasis of this lateral meristem through a positive regulatory cascade. Cambium cell proliferation is regulated by the WOX4 TF [110], which interacts with the HAM4 TF [13], and also by the WOX14 TF, which acts redundantly with WOX4 in regulating vascular cell proliferation [111]. 

As in the cases of *WUS* and *WOX5*, the expression of the *WOX4* and *WOX14* genes is regulated via the interaction of certain CLE peptides with their receptors. The balance of the cambial SCs pool is under the control of a specific small group of CLE peptides called TRACHEARY ELEMENT DIFFERENTIATION INHIBITORY FACTOR (TDIF), also known as B-group CLEs, and in *Arabidopsis* the TDIF group includes CLE41/CLE44 and CLE42 peptides [40]. TDIFs are produced in the phloem cells, and then move to the cambium SCs where they bind to their receptor called TDIF RECEPTOR/PHLOEM INTERCALATED WITH XYLEM (TDR/PXY), a member of the LRR-RLK family, and its homologs called PXY-like (PXL1 and PXL2) [40,112]. In contrast to the apical meristems, where CLEs act as negative regulators of SC maintenance, the interaction of TDIF peptides with the PXY receptor represses the differentiation of cambium SCs into secondary xylem and promotes cambium identity [113]. Another way in which *WOX4* expression is positively regulated is through IAA-dependent control by auxin response factor (ARF) TFs and class III homodomain and ZIP domain (HD-ZIPIII) TFs. These TFs are involved in xylem formation, and this pathway is essential for the establishment of cambial OC [27].

At the same time, TDIF-dependent control of the direction of cell divisions in the cambium requires the presence of a functionally active PXY receptor and is not dependent on the activity of the WOX4 TF [114]. It is proposed that TDIF signalling provides positional information to PXY to maintain the activity and bifacial nature of cambial SCs via three distinct pathways [115].

Firstly, PXY represses xylem cell differentiation by associating with the Glycogen Synthase Kinase 3 (GSK3) kinase family proteins BRASSINOSTEROID INSENSITIVE2 (BIN2), BIN2-LIKE1 (BIL1) and BIL2 at the plasma membrane [116]. This interaction in turn leads to the phosphorylation and subsequent destabilisation of the BRASSINAZOLE-RESISTANT1/BRI1-EMS-SUPPRESSOR1 (BES1/BZR1) TFs, which redundantly promote xylem differentiation [117]. Thus, PXY signalling maintains a pool of undifferentiated SCs in the cambium, but this function is independent of WOX4 [111]. 

Secondly, TDIF–PXY signalling promotes cambial SC proliferation by upregulating *WOX4* and *WOX14* gene expression, and it has been suggested that the TDIF–PXY–WOX4/14 regulatory module controlling cambial cell division acts in parallel with those regulating xylem differentiation [111].

Thirdly, TDIF–PXY influences the recruitment of cells to the phloem lineage. This is performed through a feed-forward loop involving the PXY-regulated WOX14 TF and the TARGET OF MONOPTEROS6 (TMO6) TF, which is also controlled by auxin and the MONOPTEROS TF. WOX14 and TMO6 regulate the expression of the *LATERAL ORGAN BOUNDARIES DOMAIN4* (*LBD4*) gene, which encodes a TF which is active in cells located at the procambium–phloem boundary. The *LBD4* gene governs vascular cell proliferation and phloem distribution redundantly with *LBD3* [118].

Other components of the cambial WOX-CLAVATA system in *Arabidopsis* include two close homologs of TDR/PXY receptors, PXY-LIKE (PXL1 and PXL2) [111,119]. Furthermore, PXY activation requires SOMATIC EMBRYOGENESIS RECEPTOR KINASE (SERK) co-receptors, which associate with PXY at the plasma membrane in a ligand-dependent manner [120]. In addition, two LRR-RLKs, MORE LATERAL GROWTH 1 (MOL1) and REDUCED LATERAL GROWTH 1 (RUL1), are also involved in cambium development: MOL1 is thought to be a negative regulator that reduces cambial cell proliferation independently of TDR/PXY, and RUL is a positive regulator [121].

The differentiation of phloem and xylem is also controlled by CLAVATA-like (possibly WOX-CLAVATA) systems. In particular, the formation of phloem, a tissue that produces TDIF peptides, is controlled by the CLE25 and CLE45 peptides and their receptors: CLE Receptor Kinase (CLERK)-CLV2 and BAM3, which may use CLAVATA3 INSENSITIVE RECEPTOR KINASE (CIK) proteins as co-receptors [122,123]. Several mutations in *CLE, BAM* or *CIK* class genes caused ectopic formation of phloem clusters [123]. In addition, several TFs have been found to regulate cell specification to a phloem type, e.g., the ALTERED PHLOEM DEVELOPMENT (APL) and LBD1 TFs repress the differentiation of phloem progenitor cells along the xylem pathway [124,125]. There are also several genes encoding DOF-class TFs that are preferentially expressed in the phloem (so-called phloem-DOFs). Recently, phloem-DOFs have been shown to induce the expression of negative regulators of phloem development, the CLE25, CLE26 and CLE45 secretory peptides. These CLE peptides could in turn interact with the BAM-CIK receptor complex to post-transcriptionally reduce the amount of phloem-DOF protein, thereby leading to phloem element differentiation [123]. The dual function of the CLAVATA-like system and its interaction with the phloem-DOFs has been proposed: the phloem-DOFs induce phloem cell formation while simultaneously inhibiting excessive phloem cell formation by inducing certain CLEs outside the phloem zone [123]. 

The development of xylem, a tissue whose differentiating cells transport the IAA necessary to position the cambium OC [28], is blocked by TDR-PXY signalling (see above). At the same time, specific CLE peptides that stimulate wood formation have been identified in woody species such as poplar and birch. In *Populus*, *PtLRR-RLK1* was shown to be the putative ortholog of *TDR/PXY* [126]: its overexpression led to an ectopic accumulation of lignin in the pith, together with an enlarged secondary xylem. Furthermore, in *Populus*, lateral growth is also negatively regulated by *PtrCLE20*, which moves from the developing xylem zone to the vascular cambium cells and represses their activity by inhibiting meristematic cell division [127]. In hybrid aspen, the *PttCLE47* gene is an important positive regulator of cambial activity, promoting the cell division activity of the vascular cambium in trees at its site of expression, in contrast to other previously characterised *CLE* genes expressed in the wood-forming zone [128].

### 3.4. WOX-CLAVATA Systems in Other Meristems

WOX-CLAVATA systems are thus conserved functional modules that regulate the activity of three major plant meristems, SAM, RAM and (pro)cambium, and also the specification of specific tissues derived from them. The exact composition of these systems can vary in different meristems, but the general principle of their action is quite similar in different parts of the plant body. 

Root nodules of nitrogen-fixing plants are symbiotic organs that function in the maintenance and metabolic integration of large populations of nitrogen-fixing bacteria and are formed under specific conditions using conservative mechanisms [129]. In the symbiotic nodules, a CLAVATA-like system forms the basis of the autoregulation of nodulation (AON), which provides systemic control of nodule development at the whole organism level [130]. The AON system involves the CLV1-like receptor kinase [131,132], which functions in the shoot and is closely related to the CLV1 kinase in *A. thaliana*. Homologues of genes encoding such a CLV1-like kinase have been found in *Medicago truncatula* (*MtSUNN*), *Lotus japonicus* (*LjHAR1*), *Pisum sativum* (*PsSYM29*) and *Glycine max* (*GmNARK*) [132,133]. 

In legumes, certain root-produced CLE peptides come from the roots to the shoot in response to rhizobial inoculation and bind to the CLV-like receptor. Each legume species has AON-specific CLEs, e.g., MtCLE12, MtCLE13 and MtCLE35 in *M. truncatula* [134], LjCLE-Root Signalling1 (LjCLE-RS1), LjCLE-RS2 and LjCLE-RS3 in *L. japonicus* [133,135], Rhizobia-induced CLE 1 (RIC1) and RIC2 in *G. max* and *Phaseolus vulgaris* [136], and so on. CLE genes have been described in some legume species, notably the *LjCLERS2* gene from *Lotus japonicus* [133], and *MtCLE35* from *M. truncatula* [137,138,139], whose expression is also stimulated by nitrate. CLE peptides provide a systemic signal from the roots that ‘informs’ the shoot of the nitrogen status of the soil, and such a signal can also be perceived by autoregulatory components that promote the activation of another signal from the shoot to the roots that inhibits further nodule formation [130]. One of the possible targets may be the *WOX5* gene. *WOX5* expression is induced in pericycle cells during nodule initiation and is maintained during early stages of nodule meristem development. In *M. truncatula*, the supernodulating mutant *sunn* with the loss of CLV1-like receptor function is characterised by an expansion of the expression zone for *MtWOX5* in the nodule, suggesting the involvement of SUNN LRR-RLK in restricting the expression of this gene [140].

Another regulator that may be targeted by AON is the NIN (NODULE INCEPTION) TF, which is specifically induced by rhizobia in inoculated plant roots. This key TF activates several regulatory modules during nodulation [141,142]. Its expression is regulated by the MtIPD3/LjCYCLOPS TFs, which are components of the signalling cascade triggered in response to rhizobial infection, and by the CK-induced pathway [143]. NIN directly activates the expression of AON-specific *CLE* genes, which are negative regulators of symbiotic nodule formation [137,143]. In addition to NIN, the TFs of the NLP (NIN-like proteins) family upregulate the expression of nitrate-induced *CLE* genes (*MtCLE35* in *M. truncatula*) in response to nitrate treatment [137,139].

There are also data on the role of WOX-CLAVATA systems in the formation of another highly specialised plant organ—the periderm, which acts as a protective armour on stems and roots of perennial plants, replacing the function of primary protective tissues such as the epidermis and the endodermis. The periderm consists of a LM called the phellogen, or cork cambium, and its derivatives: the parenchymatous phelloderm in the centre and the lignified and suberized phellem outwards. In the majority of plants, the phellogen originates from the subepidermal layer and always begins after the initiation of the vascular cambium [144]. Lineage tracing analysis of the phellogen in *Arabidopsis* roots revealed that this LM originates from the pericycle, which can give rise to both the phellogen and the vascular cambium [145]. The expression of certain cambium-related genes, including *WOX4*, was shown to occur in the phellogen [146,147]. Furthermore, the onset of *WOX4* expression at phellogen initiation, as in the cambium, requires high auxin levels and the activity of certain auxin-responsive ARF TFs [146]. 

In response to various biotic and abiotic stimuli, plants can develop disorganised cell masses, such as callus and tumours, which can acquire meristem characteristics and give rise to new meristems, for example, when regeneration is induced. Numerous meristem regulators, such as the PLT family of TFs, contribute to callus formation at wound sites [148] and plant regeneration [149]. Several studies have shown that callus formation on auxin-rich callus-inducing media follows a developmental programme of root formation: the forming calli resemble those of root primordia with organised expression of root meristem regulators such as WOX5 [150,151]. Moreover, WOX5 is also essential for shoot regeneration from callus [152]. More recently, other WOXs have been identified as regulators of callus formation in *Arabidopsis* and other plants. These include WOX2, WOX8 and WOX9, which are known as regulators of zygotic embryogenesis and are also essential for callus induction and regeneration [153,154], WOX13, which belongs to an ancient subclade of the WOX family and plays a key role in the of callus formation and organ adhesion [155], and WUS, which is required for the production of plant embryonic SCs during callus regeneration [55,156]. 

In *Arabidopsis*, CLE1-CLE7 peptides have been shown to mediate shoot regeneration from callus, presumably by acting through CLV1 and BAM1 receptors to regulate *WUS* expression [157]. A further three CLE peptides, MtCLE08, MtCLE16 and MtCLE18, which are possible regulators of callus formation and regeneration, have recently been identified in *M. truncatula*, of which MtCLE08 was found to inhibit expression of the *MtWOX13* gene [158]. 

Tumours in higher plants are abnormal tissue outgrowths that result from the uncontrolled proliferation of a group of cells that acquire meristem-like properties. Most plant tumours are formed under the influence of various pathogens, and spontaneous tumours, which develop in plants of a particular genotype, are much rarer [159]. One of the most studied examples of spontaneous tumours—cambium-derived tumours on the taproots of radish inbred lines—was shown to form meristematic foci that were located in the tumour periphery that resemble RAMs, including the presence of auxin response maxima and *WOX5* expression [160]. 

Among pathogen-induced tumours, the role of the WOX-CLAVATA regulatory module in the tumour induction has been most extensively studied in root galls induced by cyst and root knot nematodes. These plant pathogens produce effectors that mimic plant peptide hormones, including CLEs, which are essential for pathogenesis (reviewed in [159]). Such nematode CLEs are processed in plant cells [161], bind to plant receptors such as CLV1, CLV2/CRN, BAM1, BAM2 and RPK2 [162,163] or PXY [164], and influence plant cell differentiation and the formation of nematode feeding sites in plant roots [165,166]. One of the identified targets of nematode CLEs in plant roots is the *WOX4* gene [164]. In general, the ability to produce plant signal peptides is not a unique feature of phytoparasitic nematodes. Molecular mimicry of effectors produced by phytopathogens for peptide phytohormones of different classes is widely represented in different groups of phytopathogens (bacteria, fungi, nematodes) and has occurred several times in evolution (see review [21]).

Thus, the WOX-CLAVATA regulatory module, which is central to meristem maintenance, is highly conserved across different meristems in higher plants. They may interact with different regulators and control different genes that are targets of the WOX TF, but they have a common basis. This is likely to be evidence of their ancient evolutionary origin [167]. Furthermore, this highly conserved module, which regulates the maintenance of SCs, meristems and ultimately the production of new organs and tissues, has become one of the targets of pathogen effectors that mimic peptides [21,168].

## 4. Module 2: Florigen-Activating Complex (FAC)

The most important event in the plant life cycle is the transition to sexual reproduction. In angiosperms, this phase transition consists of the transformation of the vegetative SAM, which forms leaves at its periphery, into the inflorescence meristem, on the periphery of which floral meristems form to give rise to flowers. Such a switch in the developmental programme is controlled by a series of complex regulatory pathways that integrate information from the environment, primarily temperature and photoperiod. The WOX-CLAVATA operating in the vegetative SAM also functions in the inflorescence and floral meristems, and the termination of SC maintenance in the floral meristem is due to the epigenetic repression of the *WUS* activity in the developing flower [22]. However, the transition to flowering is under the control of another regulatory module that also regulates the phase transition in the other processes, e.g., storage organ development.

Thus, another example of a conserved functional module that controls phase transitions in plants (transition from vegetative to generative development; entry into dormancy before the onset of unfavourable conditions; development of storage organs) is the FAC [169]. The formation of FACs is likely to be conserved among seed plant species [22,170]. 

The FAC complex consists of a several elements: (1) FT-like mobile proteins of the Phosphatidylethanolamine-binding proteins (PEBP) family, which are transcriptional coregulators; (2) 14-3-3 family proteins, conservative molecular adaptors that interact with phosphorylated serine and threonine residues in eukaryotic proteins and often interact with bZIP (Basic Leucine Zipper) TFs [171]; (3) TFs of the FLOWERING LOCUS D (FD) bZIP family [172]. In *A. thaliana*, FAC formation requires phosphorylation of a threonine residue at position 282 of the FD protein, which is catalysed by the calcium-dependent protein kinases CPK6 and CPK33 [173]. Doubled molecules of these three components form a hexameric complex that binds to G-box-containing motifs (CCACGTGG) in the promoters of target genes [174]. In most cases, these target genes encode TFs of the MADS family [172] (Figure 3).

### 4.1. FACs in Flowering Control

The study of the FACs regulatory module started with the discovery of a mobile photoperiod-dependent regulator of flowering—a PEBP family protein encoded by the *FLOWERING LOCUST* (*FT*) gene in *Arabidopsis*. The FT protein, which is produced in the leaves under inductive (promoting flowering) conditions, moves along the phloem [175] to the SAM, where it activates the expression of target genes that determine the initiation of floral meristems [176]. The *FT* gene is the target of the CONSTANS (CO) TF, a key regulator of flowering. In *co* mutants, the *FT* gene expression is not enhanced with increasing daylight hours. The *ft* mutants, like the *co* mutants, are characterised by late flowering under long day (LD) conditions, and overexpression of *FT* leads to earlier flowering [177]. The existence of a protein with such a function was predicted by Chailakhyan in 1937 and named “florigen” [178]. The TFL1 protein, a closely related FT homolog in *Arabidopsis*, has the opposite function: it inhibits flowering by interfering with FT function, thus acting as an antiflorigen. Subsequently, FT-TFL1-like proteins of the PEBP family have been described in several angiosperm families, acting as activators or repressors of flowering and some other developmental processes, with activators and repressors being distinguished by the presence of certain amino acid residues in conserved positions [22,179].

The *FT-TFL1-like* genes are generally expressed in the leaves under inductive light conditions and their products move along the phloem to their site of action, where they induce the expression of specific genes leading to phase transition in plant meristems. The transport of the FT protein along the phloem from the leaves to the SAM is regulated by several proteins. The FT-INTERACTING PROTEIN 1 (FTIP1), a membrane protein located in the endoplasmic reticulum, is responsible for FT transport from companion cells to sieve elements [180]. The FT-like proteins are not the only interacting partners of FTIP: it has been shown that certain FTIP family proteins can interact with STM to facilitate its recycling to the nucleus, which is important for SC maintenance in the SAM [181]. The long-distance transport of FT along the phloem is regulated by another FT-interacting protein, the NaKR1 (SODIUM POTASSIUM ROOT DEFECTIVE 1) protein, which belongs to the group of heavy metals-associated domain (HMA) protein family [182].

The following members of the PEBP family have been characterised in *Arabidopsis*: FT (FLOWERING LOCUS T), TFL1 (TERMINAL FLOWER 1), TWIN SISTER OF FT (TSF), MOTHER OF FT AND TFL1 (MFT), BROTHER OF FT (BFT) and ARABIDOPSIS THALIANA CENTRORADIALIS (ATC). Subsequently, proteins of this family were identified from other species of seed plants, and they were grouped into three main clades: MFT-like, FT-like (FT and TSF) and TFL1-like proteins (TFL1, ATC and BFT) [183]. All of them can form FAC complexes together with the 14-3-3 and FD proteins [184].

The MFT proteins are the common ancestors of the FT-like clade. They are conserved in gymnosperms and angiosperms and are responsible for regulating seed dormancy during germination. In *Arabidopsis*, the MFT is activated under far-red light and is involved in the regulation of seed dormancy by inhibiting seed germination. This cascade also involves the ABA-induced TF ABI5 (ABA-INSENSITIVE5): the MFT binds to the regulatory sequences of *ABI5*, thereby repressing its expression through a negative feedback loop [185]. In addition, the product of MFT prevents seed germination by activating the expression of genes that mediate the response to ABA, in particular, *ABI2* [186]. 

The emergence of flowering plants during evolution was preceded by the separation of *FT/TFL1-like* genes into two clades: FT-like flowering activators (florigens) and TFL1-like flowering repressors responsible for maintaining vegetative growth of plants (antiflorigens) [187].

The FT-like proteins in *Arabidopsis* include the FT and TSF proteins, which stimulate flowering under LD [188]. The FT-FD complex activates the expression of the key regulatory genes responsible for floral meristem development, in particular, *SUPPRESSOR OF OVEREXPRESSION OF CONSTANCE 1* (*SOC1*) and *APETALA1* (*AP1*) [172]. In addition, FT and TSF have another function not related to flowering regulation: they control blue light-dependent stomatal opening [189]. Members of the second clade (TFL1, ATC and BFT) repress flowering and the transcription of floral meristem identity genes, in particular, *LEAFY* (*LFY*), *CAULIFLOWER* (*CAL*) and *AP1* [190,191]. The TFL1 protein is less mobile than FT: the *TFL1* gene is expressed in the SAM CZ, from where the TFL1 protein moves to the periphery of the SAM to repress the expression of the target genes responsible for flowering [192,193]. In addition to regulating flowering time, *TFL1* is also involved in maintaining the SAM to prevent premature flowering and, once initiated, to allow indeterminate growth of the inflorescence [194].

The amino acid sequence of the FT and TFL1 proteins is quite similar (about 60%). It has been shown that the opposite functions of these proteins in regulation flowering are determined by the certain amino acids in conserved positions. The amino acid residues that differ between the activators (FT) and repressors (TFL1 and its homologue from *Antirrhinum* CENTRORADIALIS (CEN)) of flowering correspond to the conserved positions 88/85, 113/110 and 123/120 in the protein sequences and are located within the conserved phosphatidylethanolamine-binding “pocket” [195]. Experiments in which amino acid residues in the TFL1 and CEN proteins at these positions were replaced by residues typical of the FT protein, and vice versa, reversed the effect of these proteins on the induction of flowering: for example, overexpression of such a “substituted” FT resulted in delayed flowering, as observed in plants with *TFL1* overexpression, whereas overexpression of the “substituted” *TFL1* and *CEN* resulted in earlier flowering, similarly to that observed in plants with *FT* overexpression [195]. Similar data were obtained in experiments involving the replacement of individual exons and their segments between FT-like and TFL1-like proteins, which revealed that the most critical amino acid residues for protein function are those at positions 128–145 encoded by segment B of the fourth exon [196].

Comparison of the crystal structures of the FT and TFL1 proteins revealed that their critical structural difference lies in the region of the outer loop (close to the ligand-binding pocket). This region on the protein surface determines the functional specificity of the FT and TFL1 proteins and is thought to be involved in the recruitment of hypothetical additional coactivator or corepressor proteins to the FAC complex [196].

At the same time, the high sequence similarity of the FT-like and TFL1-like proteins suggests that they may be part of the same protein complexes (with the FD TF or with 14-3-3 proteins). Apparently, florigens and antiflorigens are able to compete with each other for binding to the components of FACs, which determines their antagonistic functions. In the course of evolution, as a result of duplications and subsequent divergence, a number of plant species have acquired a wide range of *FT-TFL1-like* genes that are activated under the influence of various environmental factors. Such evolutionary acquisitions have allowed plants to use various enhanced adaptive mechanisms to cope with adverse factors [197]. 

The expression of the *FT-TFL1-like* genes is regulated by various internal and external factors. For example, in *Arabidopsis, FT* expression can be regulated by a CO-mediated photoperiod-dependent activation pathway [198], SQUAMOSA PROMOTER BINDING PROTEIN-LIKE (SPL) TFs, whose expression and transcript stability depend on plant age and different environmental conditions [199], and via the vernalisation pathway by its central regulator FLOWERING LOCUS C (FLC), a TF, which negatively controls *FT* activity [200]. 

During evolution, different plant species have developed different regulatory pathways to control the activity of florigens. For example, to promote flowering under short-day conditions in rice, an alternative pathway has evolved that controls the expression of florigens that are negatively regulated by light. The central player in this pathway is the Early heading date 1 (Ehd1) TF, which belongs to the B-type family of response regulator (RR-B) family [201,202]. The expression of *Ehd1* under LD conditions is restricted by a CCT domain protein called Grain Number, Plant Height, and Heading Date 7 (Ghd7), which interacts with the CO-like TF Hd1 [202]. Similarly, the temperature-dependent control of flowering in some plants has evolved with the involvement of proteins from the same families. In the winter form of cereals, which require vernalisation for flowering, the expression of the *FT* genes depends on the flowering repressor VRN2, encoded by an orthologue of rice *Ghd7* [203]. In biennial forms of sugar beet, which also flower under vernalisation, the expression of the *FT*-like flowering repressor is repressed by the pseudo response regulator (PRR) group protein [204]. 

The involvement of FT-like proteins and FAC complexes in flowering under short day (SD) may depend not only on alternative pathways for controlling the expression of *FT* homologs (see above), but also on alternative functions of FACs. For instance, in the *Chrysanthemum* species there are three *FT*-like genes (among which *FTL3* plays a central role in flowering induction) and one *TFL1*-like gene, *AFT*, whose product functions as an antiflorigen. In contrast to *Arabidopsis*, both florigen (*FTL3*) and antiflorigen (*AFT*) in *Chrysanthemum* are expressed in leaves depending on the photoperiod (in *Arabidopsis*, *TFL1* is not expressed in leaves but only in the SAM). From the leaves, FTL3 and AFT proteins move to the SAM and regulate the transition to flowering [176], where florigen FTL3 or antiflorigen AFT can form FAC complexes with FD-like (FDL1) TF, activating the expression of the *AP1*-like gene *AP1/FRUITFUL-like 1* (*AFL1*) [205]. A unique feature of *Chrysanthemum* is that FAC complexes are not only formed in the SAM, but also in the leaves. There, they exert a systemic regulation of flowering, as the targets of FACs in the *Chrysanthemum* leaves include *FT*-*TFL1*-like genes. Thus, in *Chrysanthemum*, FACs can regulate the expression of their own components according to the feedback regulation principle: positively in the case of “autoregulation” FTL3/FDL-*FTL3* or AFT/FDL-*AFT*, and negatively in the case of regulation AFT/FDL-*FTL3* [206].

In addition, FACs can regulate not only flowering but also several other processes in plant development, such as the regulation of axillary bud activity and the formation of storage organs.

### 4.2. FACs in the Regulation of Bud Dormancy

The process of dormant bud emergence is an important phase of vegetative plant development, but is also closely linked to the flowering process.

The development of woody plants, especially in temperate latitudes, is closely linked to the seasons and is always cyclical, with periods of active development (upward growth and thickening, branching, and flowering) being replaced by periods of dormancy when unfavourable conditions occur. The role of florigens and antiflorigens in the regulation of such a periodicity of tree development, has been demonstrated in poplar (*Populus*) species, whose FT and TFL1 homologs have opposite functions not only in the initiation of flowering but also in the processes associated with the regulation of dormancy cycles and vegetative meristem activity.

Two *FT* paralogs, *PtFT1* and *PtFT2*, have been found in poplar. The *PtFT1*, which is up-regulated under a LD, represses vegetative growth and axillary bud formation and induces flowering [207]. Two *TFL1*-like genes, *PtCEN1* and *PtCEN2*, have also been identified in poplar, and the *PtCEN1* gene represses the release of buds from dormancy and maintains axillary meristems in a vegetative state, preventing their premature development into the inflorescence meristems [208]. Thus, an increase in FT1 and a decrease in *CEN1* level promote the release of buds from dormancy and stimulates the transformation of axillary meristems into inflorescence meristems. It has been shown that the expression level of the *FT* genes increases in spring, and after flowering, *FT1* expression levels decrease, while *CEN1* expression is activated, promoting further vegetative growth of the axillary buds [209]. 

In *Arabidopsis*, two proteins with florigen activity, FT and TSF, also play a role in the floral transition in the axillary meristems. In addition to moving from the leaves to the SAM, FT and TSF also move to the “dormant” SAMs of the axillary buds. The delay in axillary bud flowering depends on BRANCHED1 (BRC1), a TCP family TF, which binds to FT and TSF without the involvement of 14-3-3 proteins and inhibits their function in axillary meristems. In the *brc1* loss-of-function mutant, which shows accelerated floral transition of the axillary shoots, such binding of florigen is impossible. At the same time, BRC1 is unable to interact with TFL1 [210].

### 4.3. FACs in the Regulation of Storage Organ Formation

In some groups of plants, FACs have acquired another specialised function during evolution: they regulate the formation of modified underground shoots—tubers and bulbs—which enable plants to survive unfavourable conditions, accumulate nutrients and are also used for vegetative reproduction. The formation of such structures depends on the length of the day, humidity and temperature [205,211].

The tuber of the potato (*Solanum tuberosum*) is a modified shoot formed during the growth of the subapical part of the stolon (underground shoot). Photoperiod also has a significant effect on tuberisation in potato, and the mechanisms underlying this regulation are similar to those underlying the regulation of flowering. In wild forms of potato, tubers are formed at SD lengths, but many modern cultivars have lost the photoperiodic control of tuberisation.

The mechanism of photoperiod-dependent tuber formation has evolved through the divergence of the FT proteins functions. In addition to the florigen SP3D, which is exclusively involved in the control of flowering, additional regulators such as tuberigen (SP6A) and antituberigen (SP5G and SP5G-like) have evolved in potato to control tuber development. The *SP6A* gene is expressed at high levels in the leaves and stolons of short-day plants; the SP6A protein is transported to the stolon, where it can induce tuberisation. Plants overexpressing *SP6A* formed tubers regardless of day length, and silencing of this gene delayed tuberisation even under inductive SD. The accumulation of *SP6A* transcripts in leaves correlates with the time of tuber formation [212]. This gene was expressed at low levels in leaves during both LD and SD [213]. The expression of *SP6A* during the SD depends on the negative regulation of its expression by the CO TF, which accumulates in the light [212,214]. Two other members of the potato *FT-like* gene family, *SP5G* and *SP5G-like*, are expressed in the non-inductive LD and are thought to be *SP6A* antagonists, called antituberigens [212]. The TF encoded by one of the three potato *CONSTANS-like genes*, *COL1*, directly activates the expression of *SP5G*, which in turn indirectly inhibits the expression of *SP6A* in potato leaves, thereby repressing tuberisation under LD conditions [215]. 

There are several other mechanisms controlling *StSP6A* expression and potato tuberisation: for instance, the small RNA *Suppressing Expression of SP6A* (*SES*), which is an upstream regulator of *StSP6A* accumulated under warm temperature [216], and a circadian clock component TIMING OF CAB EXPRESSION 1 (TOC1) interacts with StSP6A to inhibit its activity [217].

The potato StSP6A protein has been shown to interact with the St14-3-3 and StFDL1 proteins to form a tuberigen-activating complex (TAC), similar to that described for florigen. It is possible that the repressive function of the *StSP5G* antituberigen is due to its competition with the *StSP6A* tuberigen for binding to the 14-3-3 and StFDL1 proteins, similarly to FT and TFL1 in *Arabidopsis*. The StSP3D protein is also able to form a complex with the 14-3-3 and StFDL1 proteins, but the mechanism of action of this florigen in potato remains unexplored [213].

It seems logical that in tuber crops, there is competition for photosynthetic products between flower and tuber development. For example, in potato and Jerusalem artichoke (*Helianthus tuberosus*), the removal of flowers resulted in an increase in tuber yield [218,219]. The mechanism of such competition may be based on FT-TFL1-like proteins. Indeed, potato StSP5G represses the expression of *StSP3D*, thereby delaying flowering [212], and SP6A prevents flowering independently of SP3D [220]. 

FT-like proteins are also important in the development of another type of modified underground shoot, the bulb in *Liliaceae*. In onion (*Allium cepa*), the *FT1* and *FT2* genes, which are expressed in vegetative leaves and growing bulbs, are thought to function as florigens, while *FT4* acts as an antagonist and negative regulator of *FT1* [221]. Reduced expression of the major onion florigen, *FT2*, may regulate onion bulb formation on a SD [222].

The exact mechanisms of the FT-TFL1 function in the development of modified underground shoots are unknown, but in tomato stems, it has been shown that overexpression of florigen promotes radial expansion by stimulating secondary cell wall formation and vascular maturation [223]. Thus, FT-TFL1-like proteins and FAC complexes may be involved in the processes of secondary growth of the stem, and probably of the root, and thus may regulate storage root development. Indeed, the development of storage roots in at least some root crop species is dependent on the photoperiod [224]. On the other hand, several copies of *FT-TFL1*-like genes closely related to potato *SP6A* have been identified in the selected root crops, including cassava, sweet potato, radish, carrot and sugar beet [225,226], suggesting their role in storage root development.

## 5. Changes in Regulatory Modules: Nature and Man Worked Hand in Hand 

The modular principle of plant body organisation implies a high degree of plasticity under changing conditions, which is necessary for a stationary lifestyle. Developmental modularity can occur at different levels of organisation, from the whole organism and organ systems to the underlying regulatory genes.

Changes in body modules and their structure and function are often the result of mutations in the components of conserved gene modules that control development. According to the available data, such changes have occurred both in the course of plant evolution, with the complication of their body plan and the emergence of new organs and organ systems, and in the course of their domestication and selection to enhance the manifestation of economically important traits.

In our review, we have focused on only one module at the organismal level—meristems—and two molecular modules that control meristem development, maintenance and phase transition. However, plants have many other conserved molecular modules that regulate the important aspects of their development. Of particular note is the GLABRA module, which controls the development of root hairs in the root epidermis and trichomes (leaf hairs) in the shoot system, as well as flavonoid biosynthesis in various plant organs and mucilage biosynthesis in the seed coat. This module is highly conserved and includes several interacting TFs, the target gene *GLABRA2* (*GL2*), which also encodes a TF, and an upstream pathway that may include signalling peptides and their receptors [227,228]. In addition, functional molecular modules involving the regulation of target genes by conserved small RNAs and TFs that control the expression of corresponding genes [229] are widespread and play critical roles in plant development. In the opinion review by Ruprecht et al., the function of “auxin-related” module, which includes IAA transporters of the PINFORMED family and several TFs, was analysed in detail in an evolutionary context from mosses to angiosperms [230]. 

Here we review some examples of module changes in the course of plant evolution or breeding. The evolution of functional modules, both at the molecular and organismal level, has increased the diversity of life cycles, leading to more successful adaptation to changing environmental conditions and the conquest of new habitat niches. However, such plasticity has often been reduced during domestication and breeding to facilitate uniform plant growth and high productivity in different growth environments.

### 5.1. Changes in Regulatory Modules during Evolution

Molecular evolution is based on the appearance of genes and gene families, accompanied by the emergence or modification of biological pathways and morphological traits. Examples of changes in meristems and their regulatory gene modules can be traced back to the early stages of land plant evolution. 

#### 5.1.1. Evolution of Meristems as Morphological Modules

In the development of land plants, the size and duration of the diploid sporophyte and the haploid gametophyte phases differ in the two major lineages: the sporophyte is a dominant stage in the vascular plants (lycophytes, monilophytes and seed plants), whereas the gametophyte stage predominates in the bryophytes (liverworts, mosses and hornworts). At the same time, the AMs of plants from both lineages generally have a similar plan, suggesting that this module may have pre-existed in their common ancestor [231]. Thus, the molecular basis for the maintenance of the AM structural module may also be conserved in all land plants.

In the liverwort *Marchantia polymorpha*, the SC zone of the SAM consists of a single apical cell and its derivatives [232]. In monilophytes, the functions of the SAM and the RAM are also based on a single apical cell whose progeny (merophytes) give rise to all the cells that make up the meristem. In contrast, the AMs of seed plants have a zonal structure and generally contain SCs in the centre, the transition zone, periphery and the rib meristem. At the same time, the AMs of lycophytes could have either a single apical cell (in *Selaginellaceae*) or a zonal pattern with a number of apical initials (e.g., in *Lycopodiacea* and *Isoetaceae*). Moreover, the RAMs of seed plants are generally classified into closed and open types: in the closed type RAM, all initials originate from asymmetric division of the same SCs, whereas in the open type RAM, different tissues can be traced separately to discrete cell files of initial cells, i.e., initials of protoderm, ground tissues, procambium, and root cap [233]. In lycophytes, both types of RAMs also exist, but a QC-like region with very rare cell divisions has only been revealed in open-type RAMs [234,235]. 

In the evolution of meristem structures, AMs with a single apical cell were generally considered to be the primitive ancestor of AMs with multiple initials. At the same time, according to modern ideas, both types of AM structures (with single or multiple initial cells) could have evolved independently, or even single apical cell AMs could be derived secondarily from multiple initial cell AMs [234,235,236]. 

Interestingly, this difference between the structure of unicellular and multicellular AMs correlates with the density of plasmodesmata per unit of cell surface area. The SAMs and RAMs of seed plants, as well as those of *Lycopodiaceae* (the lineage with multiple initial cells), have a three-times-lower density of plasmodesmata compared to ferns and *Selaginellaceae*, which have a single apical cell [235,236].

The similarities between seed plants and certain lycophytes in the AM structure are all the more interesting because these land plant groups demonstrate the similarity in the function of the LM cambium: in particular, secondary growth from a vascular cambium is now only found in seed plants and lycophyte *Isoetes* [237]. However, the fossil record shows that it was previously much more phylogenetically widespread in the different lineages of euphyllophytes [237]. 

In the axial organs of higher plants, all primary tissues, including the procambium, are patterned in AMs, thus establishing the procambial architecture during AM development [238]. In the seed plants, the secondary cambium is formed as a continuous meristematic layer. During the evolution of seed plants, two types of secondary vascular meristems have evolved, namely the vascular cambium (which is a characteristic of gymnosperms and non-monocotyledonous angiosperms) and the monocot cambium (found in some monocotyledons). The formation of the vascular cambium involves the sectors of the procambium (fascicular cambium), sectors recruited from mature pith ray tissue (interfascicular cambium), and also the pericycle in the root. In general, the actively dividing vascular cambium produces secondary xylem inwards and secondary phloem outwards, resulting in a radially symmetrical vascular pattern. At the same time further events, including cambial cell divisions and phloem or xylem differentiation, can be very diverse and, together with the primary patterning of the procambium in the AMs, determine the architecture of the plant vascular system [237]. The monocot cambium forms outside the primary vascular bundles from the so-called primary thickening meristem, or pericycle, and produces the secondary cortex centrifugally, and the secondary ground tissue with secondary xylem and phloem, arranged in vascular bundles, centripetally [239]. However, both types of cambium are concentric and contain initials that undergo periclinal divisions and differentiate to produce secondary vascular tissues and play similar roles in the radial growth of the axial organs. 

Since the formation of morphological modules is based on molecular modules, the complications of the body plan involved the duplication of genes encoding components of molecular modules with subsequent subfunctionalisation or neofunctionalisation. Such events have occurred in the evolution of the WOX-CLAVATA and FAC modules considered in this review.

#### 5.1.2. Evolution of the WOX-CLAVATA Module

The modular regulation of SC activities in the three main meristem types includes the WOX-CLAVATA systems, whose components may vary in different meristems: the central role of WUS and CLV3 in the SAM, WOX5 and CLE40 in the RAM, and WOX4 and CLE41/44 in the vascular cambium. This regulatory scheme is typical of the seed plants, and recently, a number of data have been obtained that shed light on the appearance of this regulatory scheme in plant evolution (Figure 4). However, the role of the WOX-CLAVATA system in the evolution of meristem structure is still largely unclear. 

##### WOX TFs

The *WOX* genes are present in the genomes of all land plants [240,241,242]. Extensive analysis of *Viridiplantae* genomes and transcriptomes has revealed three ancient superfamilies of WOX proteins, named Type 1 (T1WOX, the WOX10/13/14 clade), Type 2 (T2WOX, the WOX8/9 and WOX11/12 clades) and Type 3 (T3WOX, the WUS, WOX1/6, WOX2, WOX3, WOX4 and WOX5/7 clades). Of these, the T1WOX superclade contains proteins from all divisions of the *Viridiplantae*, and the T2WOX superclade contains proteins from seed plants only and is absent from ferns and lycophytes [242]. Although previous studies have shown that some lycophyte and fern *WOX* homologs cluster with the *WOX8/9* and *WOX11/12* clades [239], Wu et al. found no syntenic region of the *T2WOX* genes in lycophyte and fern genomes and suggested that T2WOX genes may be lost in lycophytes and ferns [241]. 

According to data obtained in *Arabidopsis* and other seed plants, *T3WOX* genes play an important role in the meristem development [240,242,243], so similar functions are assumed for the proteins of this superclade in other land plants. Thus, it is generally accepted that the establishment of different meristems during land plant evolution is based on the evolution of genes of the T3WOX superclade; in particular, the evolution of SAM and RAM is associated with the WUS/WOX5 sub-branch, and the emergence of the leaf margin meristem and the vascular cambium is associated with the duplication of the ancestral gene to *WOX3* and *WOX4* [240]. Subsequent large-scale studies in different groups of land plants have only partially confirmed this.

Recent advances in whole-genome and transcriptome sequencing of land plants have facilitated comparative studies of the WOX-CLAVATA genes in different groups. In recent years, the expression and functions of the *WOX* genes have been studied in detail for several plant species, whose SAM and RAM contain a single apical cell and for species with multicellular AMs [234,244,245].

A gametophytic SAM of *M. polymorpha* can be regulated by the single WUS/WOX homolog [246]. Transcriptome sequencing using precise laser microdissection of meristem subdomains was performed for two species with unicellular SAM (the lycophyte *Selaginella moellendorffii* and the monilophyte *Equisetum arvense*) in comparison to the multicellular SAM of maize. The list of genes whose expression was similarly regulated in maize, *Selaginella* and *Equisetum* included the class I *KNOX* gene whose expression was associated with the apical cell and/or core region of the SAM, and the *HD-ZIPIII* gene, which was expressed in the periphery. At the same time, genes encoding components and partners of the WOX-CLAVATA module (*WOX*, *CLV1-like*, *BAM1-like*, *HAM1/2*) were not expressed in the core region or apical cell of *Selaginella* and *Equisetum* SAMs [244]. Similarly, no *WOX* gene expression was detected in the root of *Selaginella* or the QC-like region of *Lycopodim* [234,245].

In *Selaginella kraussiana*, the *SkWOX11C* gene, a proposed ancestor of the *T3WOX* superclade, is not expressed in the SAM, but its expression has been detected in the phloem in vascular bundles [247]. Similarly, in the model fern species *Ceratopteris richardii*, only the *T3WOX* clade gene *CrWUS-Like* (*CrWUL*) was found to be expressed during early root development and in the vasculature, but not in the SAM. Knockdown of *CrWUL* resulted in a decrease in root number and fewer phloem cells, but did not affect SAM development [247]. Furthermore, *CrWUL*, when expressed in the *Arabidopsis* cambium, rescued the normal phenotype of the *wox4* mutant, suggesting that the WOX function in the cambium is conserved in ferns and angiosperms [247]. 

In gymnosperms, functional conservation has been shown for some *WOX* genes with corresponding *Arabidopsis* genes in the embryonic patterning, meristem maintenance and lateral organ outgrowth [182,248]. Only single homologs of *WUS*/*WOX5* expressed in both SAM and RAM have been identified in *Pinus sylvestris*, *Ginkgo biloba* and *Gnetum gnemon* [249], but the genome of *Picea abies* contains separate *WUS* and *WOX5* genes expressed in both shoot and root [250]. Thus, it has been proposed that the *WUS* and *WOX5* genes arose as a consequence of a single *WUS*/*WOX5* gene duplication prior to the split between gymnosperms and angiosperms, but that their division by functional specialisation occurred only in the angiosperms [250]. However, in *G. gnemon*, the *WUS*/*WOX5* is expressed in the SAM periphery, suggesting that its function in SC maintenance is not conserved in seed plants [251]. The expression of *WOX4* in *G. gnemon*, *G. biloba* and *P. sylvestris* is clearly associated with the vascular cambium [240].

##### CLE Peptides

In angiosperms, a feedback pathway involving CLE and WOX regulates the stem cell population, at least in the AMs. The question of how this system was established during land plant evolution has been analysed using the example of the evolution of SAM zonation in different land plant groups including the gametophytic meristems of bryophytes [252]. The model suggests that a stem-cell limiting CLV3 pathway was derived from stem-cell promoting CLE pathways conserved in land plants via gene duplication in the angiosperm lineage. 

The genomes of *M. polymorpha* (liverwort) and species of *Anthoceros* (hornwort) both contain two *CLE* genes, one belonging to the *CLV3* subfamily (also called *R-CLE*, because of the most abundant amino acid in its CLE domain) and the other belonging to the *TDIF* subfamily (also called *H-CLE*). The functional analysis of these genes was carried out in the experiments involving overexpression or treatment of plants with corresponding peptides [253,254,255]. The CLV3-like MpCLE2 was shown to function as a positive regulator of the SC pool in the gametophytic SAM, inducing SC expansion leading to a multiple-branching phenotype. Thus, the effect of MpCLE2 in *M. polymorpha* is opposite to that of CLV3 in angiosperms and close to the function of CLE40 in the SAM [254,256]. At the same time, *MpCLE1*, which encodes a TDIF-like peptide, plays the role of a “functional analogue” of CLV3 in the SAM of *M. polymorpha:* it is also preferentially expressed in the SAM, and the overexpression of *MpCLE1* causes reduced thallus growth, indicating the opposite function to *MpCLE2* [253]. 

The mosses *Physcomitrium* and *Sphagnum* have nine *CLV3*-like genes, but no *TDIF*-like genes. The CLV3-like CLEs of *P. patens* are required for apical SC oblique division and also regulate the angles essential for the moss body structure [257]. 

Both *R-CLE* and *H-CLE* lineage genes are present in vascular plant genomes [253,254,256,258,259], and the *H-CLE* lineage with its narrow specialisation in cambium identity/vessel differentiation, appears to be more conserved and less abundant in different groups of vascular plants. The effect of treatment with exogenous TDIF was investigated in the lycophyte *S. kraussiana*, the fern *Adiantum aethiopicum* and the gymnosperm *G. biloba*. Only in *S. kraussiana* did the function of TDIF differ from that in the angiosperm: endogenous and exogenous TDIF did not inhibit xylem differentiation in developing shoots and rhizophores. At the same time, TDIF treatment suppressed the differentiation of procambial cells into xylem elements in *G. biloba* and *A. aethiopicum*, suggesting a conserved function of TDIF in extant euphyllophytes [259]. Comparative analysis of CLE peptide sequences in gymnosperms and angiosperms revealed that only one CLE, the CLE41/44-like TDIF, was 100% conserved between angiosperms and conifers; the expression of TDIFs in conifers is strictly phloem-timed, as in dicots, suggesting a similar regulation of secondary growth and wood formation in gymnosperms and dicots [260]. The role of the TDIF-TDR-WOX4 pathway in wood formation in gymnosperms has been indirectly confirmed by the work on adult *P. sylvestris* trees with annual rings [261].

At the same time, the expansion of *R-CLE* genes, especially *CLV3* homologs, is highly correlated with the estimated number of whole genome duplications in land plant evolution [256]. For example, the grasses have the largest number of CLV3 homologs, and in rice (*Oryza sativa*), two separate CLV3 homologs maintain vegetative and generative SAMs, respectively [262]. In *Populus*, there are several *TDIF* genes, some of which regulate the division plane of the (pro)cambium, while others promote vascular cell proliferation and co-regulate extensive wood formation [263]. In addition, the *CLE25*-like gene (*R-CLE* group) is also associated with wood formation [128].

##### LRR-RLKs

The LRR-RLKs are a very widespread family of proteins whose members function as receptors for extremely diverse ligands, in particular peptide phytohormones including CLEs. The study of the evolutionary history of the XI LRR-RLKs (LRXI) subfamily, which are signalling peptide receptors in angiosperms, revealed that the protein of this subfamily originated from bryophytes [258]. The LRXI subfamily is divided into seven clades, two of which, the TDR/PXY and BAM/CLV1 groups, are the receptors for TDIF-like and CLV3-like CLEs, respectively. These sister clades arose by duplication in the common ancestor of vascular plants and bryophytes, and subsequently, the TDR/PXY clade was lost in the moss lineage [258] along with its TDIF-like ligand [256]. In *M. polymorpha*, both genes encoding CLE receptors, *MpTDR* and *MpCLV1*, are preferentially expressed in the SAM. The loss-of-function mutation of *MpTDR* suppresses the negative effect of *MpCLE1* overexpression on thallus growth [253], while loss-of-function mutants of *MpCLV1* had suppressed growth of the meristematic region because they are insensitive to MpCLE2 [254]. These results suggest the receptor–ligand interactions for MpTDR-MpCLE1 and MpCLV1-MpCLE2.

Within the vascular plants, both lineages of CLE receptors are represented. In *S. moellendorffii*, two *TDR/PXY* genes are sister to the angiosperm *PXL1/2*, and three *BAM/CLV1* genes are closer to the angiosperm *BAM1*. The *BAM/CLV1* gene family has expanded significantly in angiosperms along with an expansion of CLV3-like peptides, suggesting extensive receptor–ligand co-evolution [258]. At the same time, the CLE–BAM pathway appears to be more ancient (possibly evolved from the MpCLE2-MpCLV1 pathway in the liverworts) and conserved in land plants, whereas the CLV3–CLV1–WUS pathway is derived from a gene duplication event in the angiosperm lineage [256]. Indeed, most angiosperm *R-CLE* genes cluster together with *CLV3*, while the cluster containing *CLE40* includes angiosperm and gymnosperm *R-CLE* genes [256,264]. The model describing the role of the evolution of both CLE40-BAM1 and CLV3-CLV1 pathways in the emergence of SAM zonation is considered in the review by Hirakawa [256].

The cofactors of the WOX-CLAVATA modules have also evolved with the land plants. The HAM family proteins are among the most important interacting partners of WOX TFs [13,72] and are widely distributed in all land plants. They are functionally conserved, with *HAM*s from bryophytes, lycophytes, ferns, gymnosperms and angiosperms being able to substitute the function of *HAM* genes in *Arabidopsis* mutants, maintain established SAMs and promote the initiation of new SC niches in axillary meristems [265].

#### 5.1.3. Evolution of the FAC Module

The next conserved module, florigen-activating complexes (FACs), is less ancient and was discovered in the fern *Anzolla* in addition to seed plants [266]. FACs are used by plants for the transition to sexual or vegetative reproduction, as well as for storage and/or dormancy [22].

Proteins of the PEBP family, which include TFL1/FT-like proteins, are common in both eukaryotes and bacteria and perform a variety of developmental functions [267]. As a result, these proteins appeared in plants earlier in evolution than the developmental processes they regulate, such as storage organ formation and flowering. Among plant PEPBs, MFT-like proteins are the most ancient branch: their genes have even been identified in moss genomes [197]. In the moss *P. patens*, *MFT-like* genes, as well as their homologues in flowering plants, may be involved in regulating the development of reproductive organs during the optimal photoperiod. The marchantia *MFT-like* gene, like its flowering plant homologues, is also involved in ABA-mediated regulation of brood-bud dormancy [268]. The *AcMFT* gene of the fern *Adiantum capillus-veneri* is expressed during vegetative development and is downregulated during the transition to sexual reproduction; moreover, its expression could be controlled by the photoperiodic regulation. Overexpression of *AcMFT* was shown to partially complement the *ft* mutation in *A. thaliana*, accelerating mutant flowering and activating *AP1* gene expression. The AcMFT protein is also able to interact with the FD protein [269].

The *FT/TFL1-like* clade of the *PEBP* gene family was first found in gymnosperms [269]. Phylogenetically, the gymnosperm-specific group of *FT/TFL1-like* genes belongs to a separate cluster located between the *FT-like* and *TFL1-like* genes of angiosperms. In *Picea abies*, two FT/TFL1-like genes, *PaFTL1* and *PaFTL2*, have been identified in addition to two *MFT-like* genes. These two genes combine the characteristics of *TFL1-like* and *FT-like* genes of flowering plants. Overexpression of *PaFTL1* and *PaFTL2* in *Arabidopsis* resulted in later flowering, whereas overexpression of *PaMFT-like* genes had no effect on flowering time. *PaFTL1* was expressed in male generative shoots, whereas *PaFTL2* expression was high in needles, as well as in vegetative and generative buds. *PaFTL2* expression increased during SD and correlated with the cessation of growth and the initiation of generative shoots in spruce. As for the *PaMFT-like* genes, they were found to be relatively highly expressed in seeds [183].

The strong functional link between the components of the control module means that changes in the individual components of the functional module cause changes in the operation of the module as a whole. For example, the opposing functions of the *Arabidopsis* FT and TFL1 proteins, which have relatively high amino acid sequence similarity, are determined by differences in the structure of the outer protein loop corresponding to segment B of the fourth *FT* exon [196], and single amino acid substitutions cause the conversion of the TFL1 protein to florigen, and FT to antiflorigen [195]. In angiosperms, FT homologs induce flowering and TFL1 homologs repress flowering [270].

The appearance of single substitutions in *FT-* and *TFL1-like* genes has led to changes in their functions. For example, tobacco (*Nicotiana tabacum*) has four *FT*-*like* genes, of which only one, *NtFT4*, acts as an inducer of flowering, while other genes repress flowering [271]. Sugar beet (*Beta vulgaris*) has two *FT* homologs, one of which, *BvFT2*, induces flowering, and the second one, *BvFT1*, has acquired repressor functions during evolution. Three amino acid substitutions have been shown to be responsible for the repressor functions of *BvFT1* [272]. FT proteins that have acquired a repressor function as a result of a single mutation have also been identified in soybean [273] and sunflower [274]. 

In addition, there is usually a functional divergence between several gene homologs present in angiosperm genomes, not only in terms of their ability to repress or stimulate flowering, but also in the control of other functions in the plant life. In *Arabidopsis*, for example, FT and TSF have been shown to perform two additional functions beyond those associated with florigen activity [188]. Firstly, they can modulate lateral shoot growth under long and short photoperiods, respectively [275]. Secondly, both FT and TSF induce the activation of H^+^-ATPases in stomatal guard cells in the response to blue light, thereby stimulating stomatal opening [189]. At the same time, BFT, the protein with “antiflorigenic” activity, is also a regulator of stress-responsive flowering and can delay the transition to flowering under high salinity by competing with FT for binding to FD [276].

### 5.2. Changes in Regulatory Modules during in the Course of Plant Breeding 

Thus, the emergence of new members of the protein families that are components of the FAC and WOX-CLAVATA modules, and new functions for these components, have been the factors that have allowed the evolution of plant diversity. In addition, a series of mutations that cause changes in the function of the components of these modules provide the basis for a wide range of adaptations of the plant life cycle to changing environmental conditions. For example, an equilibrium feedback system that regulates the operation of the module underlies the control of the SC pool in the meristems, thereby regulating the architecture of the plant body [277]. The balance of florigens and antiflorigens in leaves allows plants to decide whether to alter their developmental programme and switch to flowering or continue vegetative development. Mutations in *FT-TFL1-like* genes that have arisen during evolution have been fixed by selection during adaptation to specific environmental conditions [197].

In addition to evolutionary adaptations, the components of the two regulatory modules discussed in this review, FAC and WOX-CLAVATA, have fixed certain changes in the course of selection during plant domestication and subsequent breeding, leading to the emergence of agriculturally important traits in various crop species.

For example, in maize (*Zea mays*), mutations in genes encoding components of the WOX-CLAVATA system that regulate SAM activity are promising for use in achieving higher grain yields [278]. A mutation in the *FASCIATED EAR3* (*FEA3*) gene, which encodes a LRR-RLK CLE peptide receptor, extends the *WUS* expression zone and the size of the SAM. This affects yield-related traits, such as the number of kernel rows. Thus, weak alleles of *fea3* that increase the number of kernel rows while maintaining the normal structure of the SAM, could improve maize yield [279]. Several other mutations that increase the size of the vegetative SAM and inflorescence meristem in maize, also affect genes encoding LRR-RLK and CLE peptides. For example, *FEA2* is orthologous to *CLV2* [60], *THICK TASSEL DWARF1* (*TD1*) encodes a *CLV1* ortholog [280], and *ZmCRN*, which functions downstream of *FEA2*, is the ortholog of *CLV2* [281]. A very similar phenotype is exhibited by *compact plant2* (*ct2*) mutants. The *CT2* gene encodes a Gα protein, a subunit of the heterotrimeric G-protein complex, which has been shown to interact with FEA2 [60]. Ligands for the FEA2 receptor have been found, and these are two CLE peptides, ZmCLE7, which is close to *Arabidopsis* CLV3, and ZmFCP1, whose ortholog in *Arabidopsis* is CLE27 [279,281]. To sense signals from these two CLEs, FEA2 interacts specifically with CT2 and ZmCRN, respectively [281]. The *fcp1* mutants also have an enlarged SAM [279]. All of these mutations can be used in maize breeding to increase the yield. In particular, the CRISPR-Cas9-based engineering has allowed the creation of weak promoter alleles of the *ZmCLE7* and *ZmFCP1* genes, increasing the number of kernel rows and grain yield [282]. 

Similarly, mutations in the components of the FAC complex have played a role in the generation of economically important traits in a number of crops.

Cultivated forms of potato have acquired the ability to form tubers in LD through the appearance of allelic variants of the *CDF1* gene, which normally, unlike its *Arabidopsis* homolog, positively regulates tuberisation by activating *StSP6A* expression through inhibition of *StCO* expression. Of the four alleles of the *StCDF1* gene identified, only one was not truncated [283,284]. The presence of defective alleles in cultivated potato varieties leads to the formation of a truncated StCDF1 protein that is unable to bind to the FKF1-GI complex, is stable, and constitutively represses *CO* expression. At the same time, expression of tuberigen *StSP6A* is not repressed by the CO, and such plants form tubers under LD [283].

In cultivated tomato (*Solanum lycopersicum*), the expression of the *SELF PRUNING5* (*SP5*) gene, which normally delays flowering, is reduced due to the presence of a *cis*-regulatory element in this gene that results in earlier flowering [285,286]. Tomato also has a close homologue of the potato tuberigen, *SlSP6A*; however, in modern tomato cultivars, the *SlSP6A* is non-functional because it contains a single nucleotide insertion that causes the appearance of a premature stop codon [287].

In sunflower varieties, *HaFT1* alleles with a mutation in the third exon resulting in a frameshift (*HaFT1-D* allele) are common. The mutant HaFT1-D protein acts as a flowering repressor, causing later flowering. The *HaFT1-D* allele suppresses the activity of *HaFT4* [274] and the mutant protein encoded by the HaFT1-D allele is likely to compete with the HaFT4 protein for binding to the FD-like TF in the FAC. Another feature of cultivated sunflower varieties is a day-neutral flowering, which results from mutations in the cis-regulatory elements of the *HaFT4* gene that cause *HaFT4* expression in leaves under both LD and SD. 

In *Rosaceae* (genus *Fragaria* and *Rosa*), continuous flowering throughout the growing season (recessive trait) is associated with mutations in genes homologous to the *TFL1*: *RECURRENT BLOOMING/KOUSHIN* (*RB/KSN*) in members of the genus *Rosa* [288] and *SEASONAL FLOWERING LOCUS* (*SFL*) in members of the wild strawberry *Fragaria vesca* [289]. The *RoKSN* gene has acquired a retrotransposon insertion, leading to the loss of the function of this gene, and *FvKSN*, has acquired a deletion of two nucleotides in the first exon, leading to a shift in the reading frame and loss of function [290]. At the same time, in SD-flowering strawberry cultivars, *FvTFL1*, a flowering repressor activated by TF SOC1, plays a key role in flowering time control. During SD, the expression of *FvFT1* and *FvSOC1* decreases slowly, leading to a decrease in *FvTFL1* expression and the initiation of the flowering programme [291]. A mutation in the *FvTFL1* renders flowering independent of day length [292]. Continuous-flowering rose cultivars lack the functional *RoKSN* gene, a flowering repressor, and continuous floral meristem growth occurs throughout the growing season [290]. In pear (*Pyrus communis*) and apple (*Malus domestica*), RNA interference of *TFL1-like* genes resulted in earlier flowering [293].

## 6. Concluding Remarks

In this review, we have focused on two key regulatory modules underlying the development and maintenance of plant meristems and their phase transitions. In addition to the WOX-CLAVATA and FAC-like regulatory modules, there are many other examples of conserved molecular modules, including receptors, their ligands, TFs, small regulatory RNAs, etc., involved in different aspects of plant development. Such modules serve as “building blocks” that have been recruited to different developmental programmes during the evolution of higher plants. Thus, the “modularity” of plant organisation is manifested not only at the morphological level, but also at the molecular level, where conserved molecular modules regulate different types of plant meristems and their developmental fates, and this is the core of the evo-devo concept in plants. Targeting the function of such regulatory modules using genetic manipulation techniques opens up the possibility of obtaining plants with desired traits and improved agricultural productivity.

## Figures and Tables

**Figure 1 plants-12-03661-f001:**
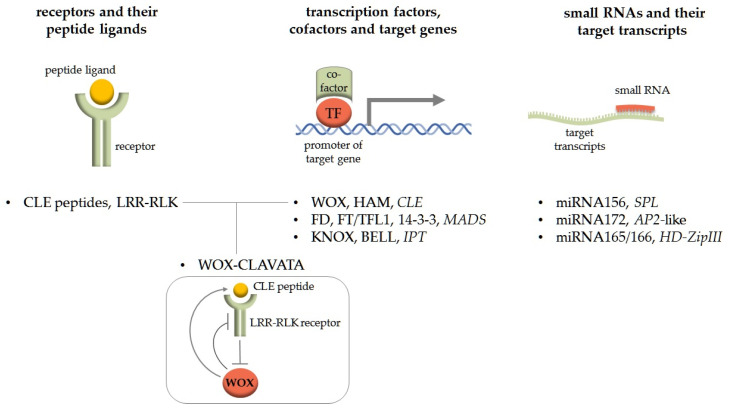
Examples of functional modules operating in meristems. A functional model could include (1) a peptide ligand and its receptor, (2) transcription factors, their cofactors and target genes, and (3) small RNAs and their target transcripts. Simpler modules can be combined into a more complex one, such as the WOX-CLAVATA module, including the mobile signalling peptides CLE, their receptors, LRR-RLKs, and the target genes encoding TFs of the WOX family, some of which could regulate the *CLE* gene expression through a feedback mechanism.

**Figure 2 plants-12-03661-f002:**
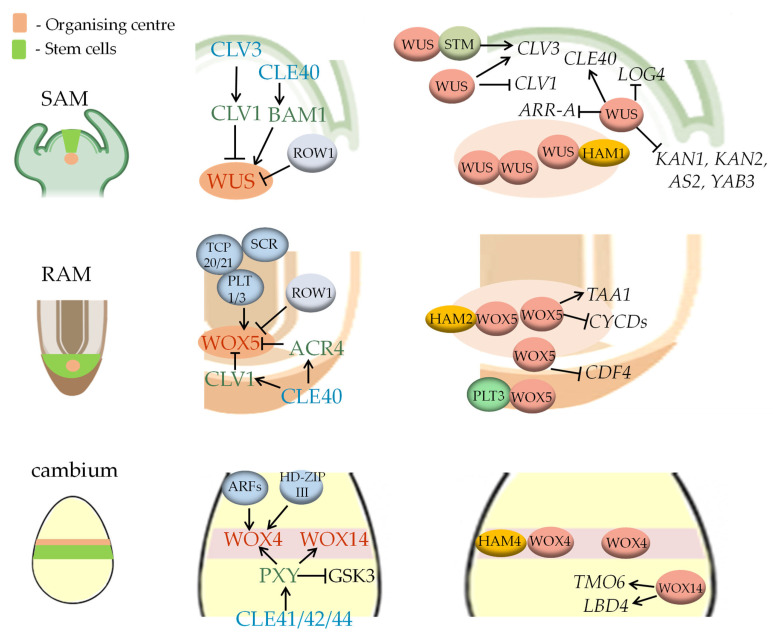
The WOX-CLAVATA module in the control of three major meristems. **Left**—zonation of meristems; **middle**—components of WOX-CLAVATA systems and other ways of regulating *WOX* expression; **right**—targets of WOX TFs. The obligate parts of the SAM, RAM and the cambium are the stem cell pool (SC) and the organising centre (OC). In the SAM and RAM, these parts represent local groups of cells, and in the cambium—cell layers. The WOX-CLAVATA system is a complex regulatory module involving the same families of proteins operating in different meristems. The WOX transcription factors (TFs), WUS in the SAM, WOX5 in the RAM, and WOX4 and WOX14 in the cambium, regulate OC maintenance and repression of SC differentiation. The expression levels of the *WOX* genes are controlled by CLE peptides and their receptors: the CLV3 and CLE40 peptides and their receptors CLV1 and BAM1 in the SAM, the CLE40 peptide and its receptors ACR4 and CLV1 in the RAM, and the CLE41/42/44 peptides and the PXY family receptors in the cambium. There are also transcriptional regulators which control *WOX* expression: ROW1 in the SAM and RAM, the complex SCR-TCP20/21-PLT1/3 in the RAM and certain ARFs and HD-ZIPIIIs in the cambium. An additional target of the PXY receptor, GSK3 kinase, was found in the cambium. The WOX TFs have different direct target genes in the meristems (see details in the text). To modulate their binding efficiency to their promoters, the WOX TFs can form homodimers or interact with other TFs: e.g., the STM TF enhances the binding of the WUS TF to the promoter of *CLV3*, while the HAM family TFs in the whole meristem and PLT3 in the RAM block the mobility of the WOX proteins.

**Figure 3 plants-12-03661-f003:**
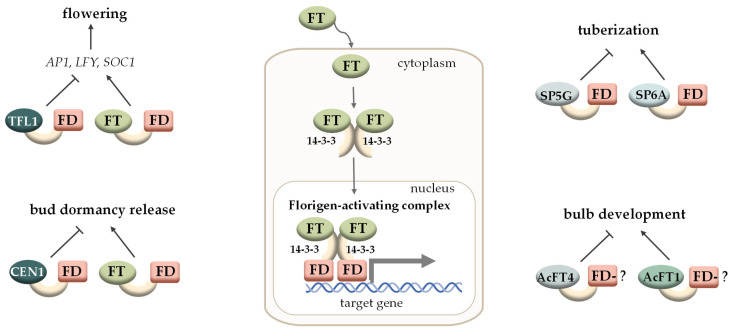
Scheme illustrating the role of alternative FAC-like complexes in the phase transitions of vegetative SAM. The FAC is a hexameric complex consisting of two molecules of FT-like mobile proteins; two molecules of 14-3-3 family proteins, conservative molecular adaptors; and two molecules of the FD transcription factor. This complex binds to G-box-containing motifs (CCACGTGG) in the promoters of target genes. During flowering induction, FT, as part of the FAC, acts as a co-activator of target genes responsible for the floral meristem development (*AP1*, *LFY*, *SOC1*). The TFL1 (“antiflorigen”) acts as a transcriptional co-repressor when it replaces FT in the FAC complex. In poplar, bud dormancy release is activated by FT in the complex with 14-3-3 and FD proteins (StFDL1), whereas CEN1, a TFL1-like poplar protein, inhibits this process. Potato tuber formation is stimulated by the FT-like protein SP6A, which forms a complex with the 14-3-3 and FD proteins (known as tuberigen-activating complex), whereas another FT-like protein, SP5G, suppresses tuber formation by inhibiting the action of SP6A. The induction of tuber formation in *Allium cepa* is stimulated by the FT-like protein AcFT1. This process is suppressed by another FT-like protein, AcFT4, and it is proposed that these proteins form alternative complexes with the 14-3-3 proteins and the FD TF and probably with other partners.

**Figure 4 plants-12-03661-f004:**
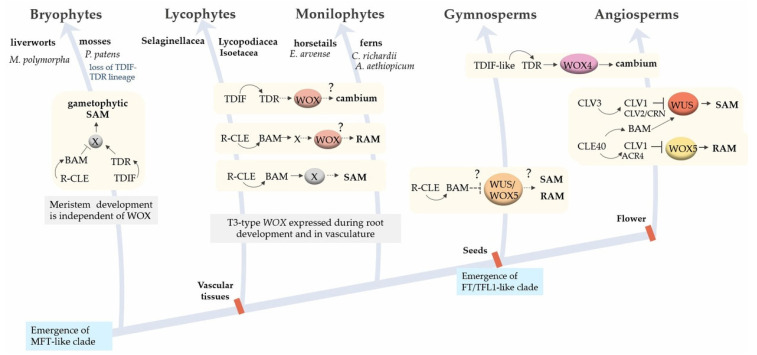
Proposed scheme for the evolutionary acquisition of the WOX-CLAVATA and FAC modules in land plants. Bryophytes: The only meristem is the unicellular gametophytic SAM. In liverworts and hornworts, two types of CLEs (TDIF and R-CLE) regulate the gametophytic SAM development in opposite directions and bind to PXY-like and BAM-like receptors, respectively. The TDIF lineage is lost in mosses. In all bryophytes, WOX do not play a role in SAM development. Lycophytes: Acquisition of vasculature and RAM; multicellular SAM and RAM in Lycopodiacea and Isoetacea. Monilophytes: unicellular SAM and RAM. In Lycophytes and Monilophytes, the TDIF-TDR pathway regulates vascular system, and R-CLEs stimulate SAM and RAM activity via PXY-like and BAM-like receptors. T3-type WOX is expressed in the RAM and vasculature, but not in the SAM. TDIF-PXY and R-CLE-BAMs are thought to be associated with T3WOX in the vasculature and RAM. Gymnosperms: Seed capture. Only multicellular SAM and RAM. T3 WOXs are expressed in the SAM, RAM and cambium; WOX4 regulates cambium, WUS/WOX5 in some species or separate WUS and WOX5 in others—SAM and RAM. TDIF-TDR signalling regulates cambium; R-CLEs stimulate SAM and RAM activity via PXY-like and BAM-like receptors. Presumably, this regulation is mediated by WUS/WOX5 pro-ortholog. Angiosperms: Flower acquisition. Only multicellular SAM and RAM. TDIF-TDR-WOX4 signalling regulates cambium, CLE40-CLV1/ACR4-WOX5—RAM; CLV3-CLV1-WUS and CLE40-BAM-X-WUS alternatively regulate SAM.

## Data Availability

Not applicable.

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
