# Peer review of "Functional Modules in the Meristems: “Tinkering” in Action"

_plants, 2023, doi:10.3390/plants12203661_

Round 1
Reviewer 1 Report
This paper is a review of what is known of the genes driving the organisation of meristems (shoot, root and cambium) across the plant kingdom, with of course a larger fraction of that knowledge coming from Arabidopsis. There are regulatory parts or "modules" (genes and interactions) that are strongly conserved across those meristems and across different plants but also a huge amount of case-specific variations. The first of those two aspects is nicely summarized in the paper's Figures 2 and 3. But that first aspect is also strengthened when the authors provide an evolutionary context in which to cover conservation and innovation (cf. a summary in Figure 4). However the second aspect, dealing with detailed exceptions, is not amenable to simple summaries or figures, and so that generates difficult reading overall.
I was happy to find a wealth of information and comparisons across organs and species collected together in this review. The drawback is that the abundance of facts hides a bit the overall picture when one reads the text. This is particularly severe in section 3 (the WOX-Clavata module). Indeed, section 4 is far easier to read, perhaps because that topic is closer to the author's work. The editor will likely agree that section 3 is so long and detailed that most readers will loose their motivation. I recommend the authors improve the readability of that section. Interestingly, I found subsection 3.4 far more readable than the previous subsections because its organization and messages are far clearer and one is less submerged by a multitude of genes. Perhaps you can be inspired by section 3.4 to modify sections 3.1, 3.2 and 3.3. Sometimes details can be presented in a table so that the text itself seems less encyclopedic.
Detailed points
- In the introduction, there is a duplication that has arisen, leading to nearly perfect repetitions, cf the two paragraphs beginning on line 129 and the two beginning on line 176.
- Lines 449-450: do you mean "division" rather than "differentiation"?
- In Figure 2, the inhibition coming out of ROW1 is missing in the top case.
- The last paragraph of section 3.3 provides a synthesis of 3.1, 3.2 and 3.3, it should not be hidden in 3.3.
- I would have the title of section 5 reflect its evolutionary content (the current title is "Discussion" which conveys little information).
- Similarly to section 3, reading of section 5.1 is "difficult".
The English is OK, but having so many facts (minor and major) provided with no hierarchy makes the reading difficult. Thus I strongly recommend the authors find ways to organize the information provided.
Author Response
Thank you very much for your review! In accordance with your comments, we corrected the found typos, renamed section 5 (Discussion), and corrected the defect in Fig. 2. We have also shortened and changed the structure of section 3.1. and 3.2., adding a table on target genes of WUS and WOX5 TFs. We couldn’t shorten the Section 5.1., but we made it more structured by breaking it into subsections. We hope that our review has become a little easier to understand for a wide range of readers.
Reviewer 2 Report
I would like to say this manuscript is a comprehensive review of the functional modules in lots of species for shoot and root meristem development. These modules include signal transduction, transcriptional regulation, and miRNAs. The organization and logic are sound and the writing is good. I only have some minor comments:
1. line 1505, "by the CO TF‘’, CO is a TF, but why do the authors want to emphasize TF here?
2. Please check if there are some double spaces between some words, such as on lines 1252 and 1276.
3. There are a few typos. for example: line 13, focuses should be focus;
Just check carefully to avoid typos.
Author Response
Thank you very much for your review! In accordance with your comments, we corrected the found typos.
Reviewer 3 Report
Excellent review! Beautifully illustrated.
Author Response
Thank you for your favorable review of our manuscript!
Round 2
Reviewer 1 Report
In this second round, I found the reading of the manuscript far smoother. I had less difficult keeping my motivation and the content has less redundancy.
Detailed points
- In the introduction, there is still a near-perfect duplication, cf sentence starting with "Loss of OC function,...".
- In Fig 1, could you include the inhibition from WOX to receptor.
- You can still improve a bit the English, you often have missing articles, incorrect plurals, and some incomplete sentences like on line 1231.
See above.
Author Response
Dear reviewer,
Thank you for your feedback on our revised manuscript. In the second round of corrections, in accordance with your comments, we removed the repeated phrase in the Introduction, added information to Figure 1 and improved the English.
Sincerely,
authors
